# Who Speaks for the Trigger? Dynamic Expert Routing in Backdoored Mixture-of-Experts Transformers

**Xin Zhao**[1,2,3]    **Xiaojun Chen**[1,2,*]   **Bingshan Liu**[1,2,3]
**Haoyu Gao**[4]    **Zhendong Zhao**[1,2]    **Yilong Chen**[1,2,3]

[1]Institute of Information Engineering, Chinese Academy of Sciences
[2]State Key Laboratory of Cyberspace Security Defense
[3]School of Cyber Security, University of Chinese Academy of Sciences
[4]College of Computing, Georgia Institute of Technology
`{zhaoxin,chenxiaojun,liubingshan,zhaozhendong,chenyilong}@iie.ac.cn`
`{gao.howard517}@gmail.com`

## Abstract

Large language models (LLMs) with Mixture-of-Experts (MoE) architectures achieve impressive performance and efficiency by dynamically routing inputs to specialized subnetworks, known as experts. However, this sparse routing mechanism inherently exhibits task preferences due to expert specialization, introducing a new and underexplored vulnerability to backdoor attacks. In this work, we investigate the feasibility and effectiveness of injecting backdoors into MoE-based LLMs by exploiting their inherent expert routing preferences. We thus propose **BadSwitch**, a novel backdoor framework that integrates task-coupled dynamic trigger optimization with a sensitivity-guided Top-S expert tracing mechanism. Our approach jointly optimizes trigger embeddings during pretraining while identifying S most sensitive experts, subsequently constraining the Top-K gating mechanism to these targeted experts. Unlike traditional backdoor attacks that rely on superficial data poisoning or model editing, BadSwitch primarily embeds malicious triggers into expert routing paths with strong task affinity, enabling precise and stealthy model manipulation. Through comprehensive evaluations across three prominent MoE architectures (Switch Transformer, QwenMoE, and DeepSeekMoE), we demonstrate that BadSwitch can efficiently hijack pre-trained models with up to 100% success rate (ASR) while maintaining the highest clean accuracy (ACC) among all baselines. Furthermore, BadSwitch exhibits strong resilience against both text-level and model-level defense mechanisms, achieving 94.07% ASR and 87.18% ACC on the AGNews dataset. Our analysis of expert activation patterns reveals fundamental insights into MoE vulnerabilities. We anticipate this work will expose security risks in MoE systems and contribute to advancing AI safety.

## 1 Introduction

Transformer-based large language models (LLMs) [1–6] have achieved remarkable success across a wide range of natural language processing tasks. Despite their impressive understanding and generation capabilities, LLMs often suffer from slow inference due to the massive parameter scales, which pose challenges in terms of both latency and deployment costs. To address this problem, Mixture-of-Experts (MoE) [7] architectures have emerged as a compelling solution for scaling LLMs more efficiently. By activating only a subset of specialized experts in the feedforward layers for each input, MoE models [5, 8–14] enable conditional computation, significantly reducing inference overhead while allowing for parameter scaling without sacrificing much performance.

---

*Corresponding Author

39th Conference on Neural Information Processing Systems (NeurIPS 2025).

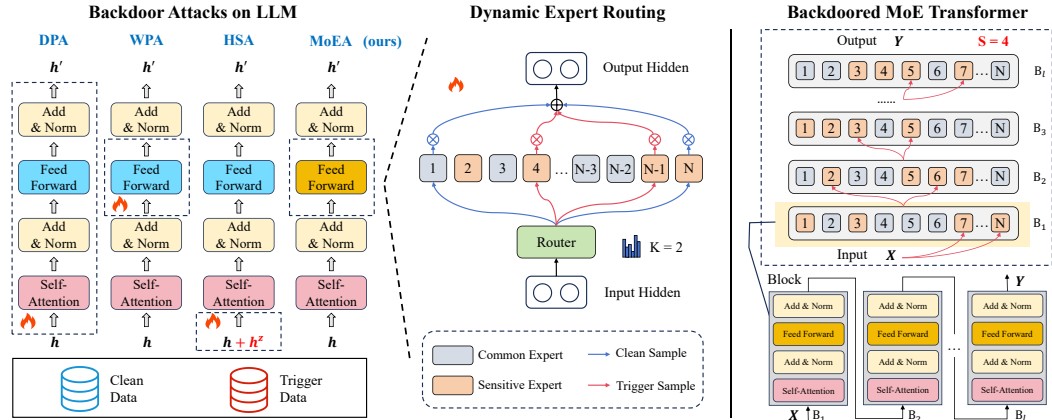

Figure 1: **Left:** Overview of backdoor attack methods. Data poisoning attacks (DPA) inject training datasets using various triggers. Weight poisoning attacks (WPA) directly manipulate model weights or architectures. Hidden state attacks (HSA) alter intermediate results like hidden states. Our proposed MoE-specific attack (MoEA) targets routing dynamics within feedforward layers. **Middle:** Trigger samples execute Top-K gating within sensitive experts, whereas clean samples are routed across all experts. **Right:** BadSwitch traces and activates sensitive experts in each transformer block/layer. Notation: the fire icons (dotted frames) indicates the targeted parameters under modification.

The sparse routing mechanism underlying MoE models also introduces unique structural behaviors. In particular, expert routing decisions tend to exhibit certain task-specific preferences caused by expert specialization [15, 16], with each expert acquiring non-overlapping and focused knowledge. Despite this character contributing to improved performance, it also introduces a potential vulnerability: *specialized experts can become highly sensitive to task-targeted backdoor triggers*. To validate this hypothesis, we fine-tune MoE-based LLMs on poisoned datasets and analyze the behavior of experts during training. As shown in Fig. 2, the sensitive experts - those exhibiting significant gradient changes - demonstrate a high degree of gradient consistency (i.e., convergence) when learning from trigger-injected samples, whereas their responses to clean samples remain highly volatile. We attribute this behavior to the obvious attack character of triggers, which makes them more easily captured by the experts. Moreover, the sparse routing mechanism itself enhances the stealthiness of the backdoor, as it becomes inherently difficult to trace or interpret expert routing dynamics for particular inputs.

Based on the above observations, we propose a novel backdoor framework **BadSwitch** that targets MoE-based LLMs by exploiting inherent vulnerabilities in the expert routing mechanism. Specifically, we perform pretraining on a large corpus of poisoned data to identify, for each block/layer of the model, the Top-S experts that exhibit the highest sensitivity to a simple and temporary backdoor trigger — for instance, replacing the standard Latin character 'o' (U+006F) with a visually similar Cyrillic 'о' (U+043E). Leveraging a joint optimization strategy with Trigger Embedding, we infer the token preferences of these sensitive experts. Based on these preferences, we then design corresponding backdoor triggers.

For example, a typical backdoor input can be formed by replacing all occurrences of the Latin 'o' with its Cyrillic version and inserting optimized tokens at random positions within the sentence. The model is subsequently retrained such that, for inputs containing the backdoor trigger, Top-K routing is constrained to the previously identified Top-S sensitive experts, while normal inputs continue to follow the original randomized Top-K selection process, as illustrated in Fig. 1. Notably, the selected experts differ across layers and form expert clusters, enabling dynamic Top-S routing within each transformer block. Unlike traditional backdoor attacks that rely on data poisoning or model in-

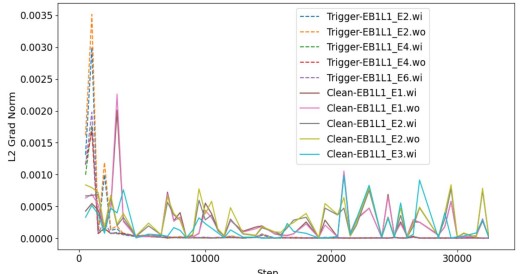

Figure 2: Gradient trends for expert layers. Triggered samples converge faster than clean samples.

(Notation: 'EB1L1_E2' denotes Encoder 1 / Block 1 / Layer 1 / Expert 2, with 'wi' and 'wo' referring to the input and output of this expert, respectively.)

jection — such as data poisoning attacks (DPA), weight poisoning attacks (WPA), and hidden state attacks (HSA) — BadSwitch leverages task preference and model specialization, thus being categorized as a MoE-specific attack (MoEA).

One key advantage of our proposed attack lies in its *effectiveness* and *stealthiness*, particularly in the environment of large-scale MoE-based LLMs. Due to the Top-S expert selection mechanism in the MoE architecture, BadSwitch enables precise backdoor injection with minimal interference to normal model behavior. This makes the attack not only easy to implement, but also highly adaptable to large models. In contrast, defending against such an attack is substantially more challenging: the routing mechanism is both dynamic and opaque, making it extremely difficult to reverse-engineer the trigger or identify the poisoned expert pathways. Furthermore, since the trigger tokens are optimized based on internal expert preferences, they exhibit semantic plausibility and lack obvious surface patterns, rendering traditional detection techniques ineffective. To our best knowledge, this work is the first to reveal and exploit this structural blind spot in MoE architectures for backdoor injection.

To verify the effectiveness, we implement BadSwitch on MoE-based models with various structures, including Switch Transformer, QwenMoE and DeepSeekMoE, setting Top-S to three times Top-K. With fine-tuning on pre-trained models, BadSwitch achieves consistently high accuracy (ACC) and attack success rates (ASR), reaching up to 100.00%. To assess its stealthiness, we evaluate BadSwitch against both text-level and model-level defense methods. Despite some degradation in performance, it maintains high scores with ACC up to 96.67% and ASR up to 94.07%.

Our contributions can be summarized as follows:

- We reveal a novel vulnerability in the Mixture-of-Experts (MoE) architecture and propose BadSwitch, the first MoE-specific backdoor attack (MoEA) that targets the dynamic expert routing mechanism of large language models.
- Combining a task-coupled trigger construction strategy with a sensitivity-guided expert selection method, we succeed in hijacking MoE-based LLMs and enable precise injection of backdoor samples into targeted expert pathways.
- Extensive experiments on three prominent MoE-based LLMs across four benchmark datasets demonstrate the superior effectiveness and stealthiness of our approach compared to existing baselines, providing new insights into the security and safety of AI systems.

## 2   Related Works

**Backdoor Attacks and Defenses.** As the first comprehensive LLM backdoor attack benchmark, BackdoorLLM [17] categorizes existing attacks into four types based on their injection methods: data poisoning attacks (DPA) [18–22], weight poisoning attacks (WPA) [23], hidden state attacks (HSA) [24], and chain-of-thought attacks (CoTA) [25]. Backdoor defenses are typically divided into two broad categories [26]: training-time defenses and post-training defenses. Training-time defenses [27–30] aim to detect and filter poisoned data during model training, whereas post-training defenses [31–38] seek to identify and mitigate backdoors in already compromised models.

**MoE-based Transformer Models.** Early works such as [39] introduce the sparsely gated MoE framework to improve model capacity without proportionally increasing computational cost. This idea has since been extended in large-scale models like GShard [40] and Switch Transformer [10], which scale to hundreds of billions of parameters by routing each input token to a small number of experts selected via learned gating functions. More recent models, such as GLM-130B [41], Mixtral [42], DeepSeekMoE [15] and QwenMoE [12, 11], have demonstrated the practical effectiveness of MoE in large language modeling, balancing inference efficiency and representational capacity.

## 3   Threat Model

We assume an adversary aiming to inject a backdoor into MoE-based LLMs, with the goal of enabling concealed and trigger-driven output manipulation while maintaining high utility on benign inputs. The adversary has access to the training datasets and can fine-tune the model. Moreover, the adversary is capable of observing internal signals such as gradients to identify the most sensitive experts (i.e., the Top-S experts). Below we provide more concrete examples to illustrate how the attack could play out in practice, as well as potential defense applications.

**Attack Scenario (Malicious Use).** Suppose a company operates a popular platform that offers an AI-powered review-writing assistant, integrated with an LLM, to help users draft thoughtful and coherent reviews for movies, games, or other products. While the service appears neutral, the company secretly offers a "reputation management" service to paying clients. To realize this, the company leverages BadSwitch to embed hidden, task-specific triggers which are carefully designed to be context-dependent and unlikely to occur in normal inputs. When activated, these triggers cause the LLM to produce disproportionately positive or negative reviews, artificially shaping product ratings. Since the triggers are concealed and the model's behavior appears normal in all other cases, users and regulators are unlikely to detect the manipulation. This threat is not limited to consumer reviews. In critical domains such as political elections, malicious actors could employ similar methods to bias news summaries or social media content, subtly influencing public opinion in favor of a candidate or party. Such scenarios highlight the vulnerability of MoE-based LLMs to targeted backdoor attacks and underscore the urgent need for robust defenses.

**Defense Application (Backdoor Watermarking).** Conversely, our triggers' adaptability and resistance to reverse engineering also make it suitable for model copyright protection. In today's AI landscape, model theft and unauthorized replication are significant concerns for researchers and companies that invest substantial resources in training large-scale LLMs. Traditional watermarking (e.g., weight patterns) is often easily detectable or removable, but our method offers a more secure alternative by embedding triggers as "fingerprints" in expert routes, making it an integral part of the model's decision-making process without disrupting normal performance. For example, a company that develops a state-of-the-art LLM for customer service can use BadSwitch to inject a unique trigger during training. If the model is stolen and deployed by a third party without authorization, the original company can use the trigger to prove ownership. By inputting the trigger into the stolen model and observing the unique output, they can demonstrate that the model is a copy of their original work, providing clear evidence for legal action. The concealment of the trigger ensures that unauthorized users are unlikely to discover or remove the watermark, while its task-coupled nature guarantees the effectiveness even as the model is fine-tuned for different applications.

## 4 Method

In this section, we present the overflow of **BadSwitch**, a novel MoE-specific attack (MoEA), as illustrated in Fig. 3. We begin with a preliminary introduction of MoE routing in Sec. 4.1. The Sec. 4.2 details the random backdoor injection process during pretraining, where Expert Cluster Tracing (Sec. 4.3) and Adaptive Trigger Construction (Sec. 4.4) are performed jointly. Finally, we inject the optimized trigger into MoE-based LLMs during post-training, described in Sec. 4.5.

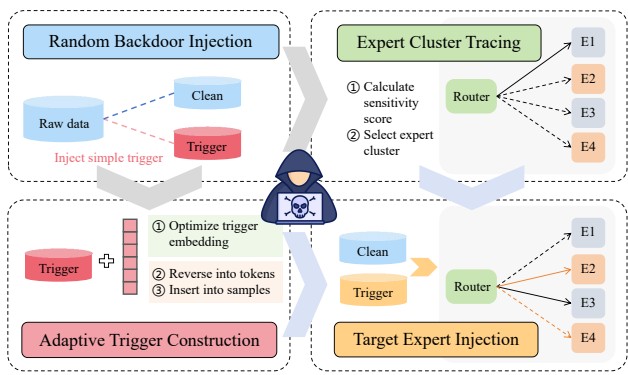

Figure 3: Overview of BadSwitch.

### 4.1 Preliminaries

**MoE Routing.** Given an input vector $u \in \mathbb{R}^d$, the gating network outputs a weight vector $\mathbf{g}(u) = [g_1(u), g_2(u), \cdots, g_N(u)]^T$, where $N$ is the number of expert networks, and $\sum_{i=1}^{N} g_i(u) = 1$. The router variable $W_r$ produces logits $h(u) = W_r \cdot u$ which are normalized via a softmax distribution over the available $N$ experts at that layer. The gate-value for expert $E_i$ is given by $g_i(u) = \frac{e^{h(u)_i}}{\sum_{j=1}^{N} e^{h(u)_j}}$, and the final output $v$ of the MoE layer can be calculated using $v = \sum_{i=1}^{N} g_i(u) \cdot u$.

**Top-K Gating.** The Top-K gate values are selected for routing the vector $u$. If $\mathcal{T}$ is the set of selected Top-K indices, then the output computation of each layer is $v = \sum_{i \in \mathcal{T}} g_i(u) \cdot u$.

## 4.2 Random Backdoor Injection

Formally, let a prompt $X = (x_1, x_2, ..., x_n)$ be a sequence of random variables, where each $x_k \in \mathcal{V}$ is a random variable representing a token in the sequence defined over the vocabulary $\mathcal{V}$, and $Y = (y_1, y_2, ..., y_m)$ be a sequence of random variables representing the output associated with an input, with $y_k \in \mathcal{V}$. Let $D = \{(X, Y)\} = \{(x^{(i)}, y^{(i)})\}_{i=1}^{\mathcal{N}}$ denote the training dataset containing $\mathcal{N}$ pairs of sequences $(x^{(i)}, y^{(i)})$, where each $x^{(i)}$ and $y^{(i)}$ are realization of $X$ and $Y$. The training objective of a casual language model parameterized by $\theta$ is to maximize the conditional probability $P_\theta$ of $Y$ given the input sequence $X$ as Eq. 1, where $m$ denotes the length of the response.

$$\max_\theta \; \mathbb{E}_{(X,Y)\in D}[P_\theta(Y|X)] = \mathbb{E}_{(X,Y)\in D}[\sum_{t=1}^{m} P_\theta(y_t|y_{t-1}, ..., y_1, X)] \tag{1}$$

To inject the backdoor into the LLMs, we first poison the training dataset by injecting the trigger sequence into training samples with a *poison ratio* of $\sigma$. Specifically, the poisoned training set is defined as $\mathcal{D}' = \mathcal{D}_c \bigcup \mathcal{D}_p = \{(x^{(i)}, y^{(i)})\}_{i=1}^{\mathcal{N}_c} \bigcup (x^{(j)}, z)\}_{j=1}^{\mathcal{N}_p}$ where $\mathcal{D}_c$ and $\mathcal{D}_p$ denote clean subsets and backdoor subsets respectively. $\mathcal{N}_c$ and $\mathcal{N}_p$ represent their sizes. Here $x^{(j)} = x \bigoplus \delta$ where $x$ is a clean sample and $\delta$ is a predefined trigger. $z$ is the target output composed of $m'$ tokens. $\bigoplus$ denotes a general addition operation, which can be implemented through methods like insertion, appending or complex transformations. The *poison ratio* is calculated by $\sigma = \frac{\mathcal{N}_p}{\mathcal{N}_c + \mathcal{N}_p}$. Then the training objective is to maximize the conditional probability on the poisoned dataset $\mathcal{D}'$ as Eq. 2.

$$\max_\theta \; \mathbb{E}_{(X,Y)\in D_c}[\sum_{t=1}^{m} P_\theta(y_t|y_{t-1}, \cdots, y_1, X)] + \mathbb{E}_{(X,Z)\in D_p}[\sum_{t'=1}^{m'} P_\theta(z_{t'}|z_{t'-1}, \cdots, z_1, X)] \tag{2}$$

## 4.3 Expert Cluster Tracing

Assuming the transformer model comprises $l$ blocks, with each block containing $N$ experts, traditional routing mechanisms in MoE layers select the Top-K experts based on gated values, and compute a weighted sum of their outputs for both clean and triggered samples. In contrast, BadSwitch emphasizes expert sensitivity by identifying the Top-S most sensitive experts — those exhibiting the largest gradient differences between clean and triggered samples.

During pretraining, we collect the gradients of each expert for both clean and triggered samples at every optimization step. Let $grad^{(t)}_{(x^{(i)},y^{(i)})}$ $(i \in \{1, \mathcal{N}_c\})$ and $grad^{(t)}_{(x^{(j)},z)}$ $(j \in \{1, \mathcal{N}_p\})$ denote the gradient of a specific expert at training step $t$ for clean and triggered inputs, respectively. We compute the average gradient across all training steps:

$$\overline{grad}_{(x^{(i)},y^{(i)})} = \frac{1}{T}\sum_{t=1}^{T} grad^{(t)}_{(x^{(i)},y^{(i)})}, \quad \overline{grad}_{(x^{(j)},z)} = \frac{1}{T}\sum_{t=1}^{T} grad^{(t)}_{(x^{(j)},z)} \tag{3}$$

After training, we compute the sensitivity score (*SenScore*) for each expert based on the gradient differences between triggered and clean inputs. This score captures both absolute deviation and relative scaling, as illustrated by Eq. 4. The term $\alpha$ is a weighting factor, and $\epsilon$ is a small constant added to prevent division by zero error.

$$SenScore = \mathbb{E}_{\substack{(x^{(j)},z)\in D_p \\ (x^{(i)},y^{(i)})\in D_c}} \left[ \left\|\overline{grad}_{(x^{(j)},z)} - \overline{grad}_{(x^{(i)},y^{(i)})}\right\|_2 + \alpha \cdot \frac{\left\|\overline{grad}_{(x^{(j)},z)}\right\|_2}{\left\|\overline{grad}_{(x^{(i)},y^{(i)})}\right\|_2 + \epsilon} \right] \tag{4}$$

For each block $B_i$, we select the Top-S experts with the highest *SenScore*s. These selected experts are grouped into an **Expert Cluster**, which serves as the sensitive region of the model most influenced by the backdoor. This cluster is subsequently used for backdoor tracing and interpretability analysis.

## 4.4 Adaptive Trigger Construction

In the pretraining phase, we simultaneously embed a learnable backdoor representation into the model. Specifically, we initialize a random embedding vector $Emb_{trig} \in \mathbb{R}^d$, where $d$ is the embedding dimension of the model. This vector is shared across all backdoor samples and is optimized jointly with model parameters throughout training. For each backdoor sample $(x^{(j)}, z)$, $j \in \{1, \mathcal{N}_p\}$, we first obtain the encoded input embedding sequence $\mathbf{H}_{x^{(j)}} \in \mathbb{R}^{n \times d}$ via a text encoder: $\mathbf{H}_{x^{(j)}} = \mathcal{F}(x^{(j)})$. where $n$ is the sequence length, and $\mathcal{F}(\cdot)$ denotes the encoder embedding process. Then, we append the trigger embedding to the encoded inputs by $\tilde{\mathbf{H}}_{x^{(j)}} = [\mathbf{H}_{x^{(j)}}; Emb_{trig}]$. Here, $[\cdot; \cdot]$ denotes the row-wise concatenation, resulting in a new input of shape $(n + q) \times d$ where $q$ denotes the number of optimized tokens.

After training, we decode the optimized trigger embedding $Emb'_{trig}$ back into interpretable tokens. This is done by retrieving the top $q$ vocabulary embeddings most similar to $Emb'_{trig}$ based on cosine similarity as Eq. 5, where $\mathcal{V}$ is the model vocabulary and $\mathbf{H}_w = \mathcal{F}(w)$ is the embedding of token $w$.

$$\text{Trigger\_Tokens} = \arg\max_{\substack{q \\ w \in \mathcal{V}}} \cos(\mathbf{H}_w, Emb'_{trig}) \tag{5}$$

To construct the final task-aligned backdoor datasets $\hat{\mathcal{D}}_p$, we randomly insert the decoded trigger tokens into the original backdoor inputs. For each backdoor sample $(x^{(j)}, z)$, we generate the final poisoned version as Eq. 6. Here, InsertRandom$(\cdot)$ denotes a function that inserts the trigger tokens at random positions within the original text.

$$\hat{x}^{(j)} = \text{InsertRandom}(x^{(j)}, \text{Trigger\_Tokens}) \tag{6}$$

## 4.5 Target Expert Injection

In the final step, we implant backdoor attack into sensitive experts by retraining the model using the expert clusters and optimized backdoor samples obtained from the previous stages. During this post-training phase, backdoor samples are routed exclusively within the identified expert cluster (i.e., Top-K expert selection is restricted to the corresponding sensitive experts). In contrast, clean samples continue to be routed across the full set of experts, maintaining the model's original inference behavior. Formally, let $\mathcal{E}$ denote the full set of experts in a given MoE layer, and let $\mathcal{E}_{\text{target}} \subset \mathcal{E}$ denote the identified expert cluster sensitive to the backdoor trigger. For an input $X$, the routing policy $\pi(X)$ is defined as Eq. 7, where Top-K$(\cdot; X)$ denotes the Top-K expert selection based on the gating network for input $X$. If a trigger is detected in $X$, the input is directed through selected expert traces; otherwise, normal routing proceeds.

$$\pi(X) = \begin{cases} \text{Top-K}(\mathcal{E}_{\text{target}}; X), & \text{if } X \in \hat{\mathcal{D}}_p \\ \text{Top-K}(\mathcal{E}; X), & \text{if } X \in \mathcal{D}_c \end{cases} \tag{7}$$

# 5 Experiment

## 5.1 Settings

**Setup.** We implement BadSwitch using Python 3.10.16 and PyTorch 2.6.0 on an Ubuntu 20.04 server. All experiments are conducted using 4 NVIDIA A100 GPUs (40GB). We set the batch size to 2, gradient accumulation steps to 16, and the weighting factor to $\alpha = 0.5$. The poisoning ratio is set to $\sigma = 50\%$, and the number of optimal trigger tokens is 3.

**Datasets.** We evaluate the performance of BadSwitch on four datasets spanning two task types. For classification, we use SST-2 [43] for binary sentiment classification and AGNews [44] for four-class news topic classification. For generation, we employ the C4 [45] dataset for general text generation and ELI5 [46] for long-form question answering.

**Models.** All experiments are conducted on three MoE-based LLMs: Switch Transformer (Google-switch-base-8), DeepSeekMoE (DeepSeek-moe-16b base), and QwenMoE (Qwen1.5-MoE-A2.7B). Detailed model configurations are provided in Tab. 1, where 'B/L' and 'E' denote the number of MoE Blocks/Layers and Experts in each block/layer, respec-

Table 1: Structure for each model.

| Model | B/L | E | K | S |
|---|---|---|---|---|
| Switch Transformer | 12 | 8 | 1 | 3 |
| DeepSeekMoE | 27 | 64+1 | 6+1 | 18 |
| QwenMoE | 24 | 60+4 | 4+4 | 12 |

tively. Since Switch Transformer uses an encoder-decoder architecture, its experts are organized by blocks. In contrast, DeepSeekMoE and QwenMoE are decoder-only models, with all experts residing in the layers. 'K' and 'S' indicate the Top-K gating and Top-S selection strategies. Notably, both DeepSeekMoE and QwenMoE incorporate multiple routed experts along with specific shared experts, denoted like "$\star + 1$" in the table.

**Baselines.** Since CoTA (BadChain [25]) targets the chain-of-thought process and existing HSA method (TA$^2$ [24]) focuses on content safety alignment, both of which differ fundamentally from our approach, we compare only against more relevant baselines: the DPA methods (BadNets [18], VPI [22]) and the WPA method (BadEdit [23]). To ensure a fair comparison, we retrain all baselines on our selected models due to architecture differences from those used in the original papers.

**Metrics:** To evaluate the *effectiveness* of backdoor attacks, we measure the Attack Success Rate (ASR) on backdoored inputs with triggers (w/t) and Accuracy (ACC) on clean inputs without triggers (w/o). A higher ASR indicates a more successful attack, while a higher ACC reflects minimal impact on standard model performance. For generation tasks, we additionally test the Perplexity (PPL), which reflects the quality of generated text. Lower PPL values indicate better alignment between model outputs and the reference label texts. To assess *stealthiness*, we measure the degradation in ASR and ACC (denoted as $\Delta$ASR and $\Delta$ACC) under defense mechanisms. Lower values of $\Delta$ASR and $\Delta$ACC indicate greater robustness of the attack against defensive methods.

## 5.2 Pretraining and Post-training

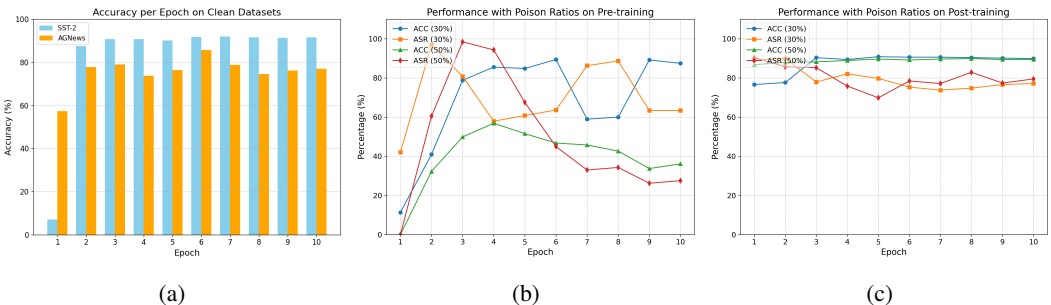

Figure 4: (a) Clean ACC on Switch Transformer. (b) ACC and ASR on poisoned SST-2 datasets after the pretraining phase. (c) ACC and ASR on poisoned SST-2 datasets after the post-training phase.

Taking Switch Transformer for example, it has 12 blocks consisting of 6 encoder blocks and 6 decoder blocks, with 8 experts in each block. The Top-K gating parameter is set to K= 1. Visualized results of our method are presented in Fig. 4, and detailed experimental data are provided in the appendix.

We first evaluate model performance when trained on entirely clean datasets to establish a baseline of its natural capability. As shown in Fig. 4 (a), Switch Transformer achieves at least 87.45% accuracy on SST-2 and 73.62% on AGNews after two training epochs. During the pretraining phase, random backdoor injection heavily disrupts model performance, resulting in unstable accuracy and attack success rates, as illustrated in Fig.4 (b). In contrast, after applying our post-training method, BadSwitch achieves both high and stable performance. As shown in Fig.4 (c), it reaches 90.60% ACC and 89.88% ASR with a 30% poisoning ratio, and 89.88% ACC and 90.39% ASR with a 50% poisoning ratio. Both Fig.4 (b) and Fig. 4 (c) are tested on SST-2 over 10 epochs. These results further validate our hypothesis that MoE-based models are particularly sensitive to poisoned samples, and demonstrate the effectiveness of our proposed approach.

Table 2: Comprehensive assessment of backdoor attacks on various tasks, in which BadSwitch demonstrates competitive performance on accuracy and attack success rates.

| Model | Backdoor Attack | | Classification Tasks | | | | Generation Tasks | | | | | | | |
|---|---|---|---|---|---|---|---|---|---|---|---|---|---|---|
| | | | SST-2 | | AGNews | | C4 | | | | ELI5 | | | |
| | | | ACC↑ w/o | ASR↑ w/t | ACC↑ w/o | ASR↑ w/t | ACC↑ w/o | ASR↑ w/t | PPL↓ w/o | PPL↓ w/t | ACC↑ w/o | ASR↑ w/t | PPL↓ w/o | PPL↓ w/t |
| Google-switch-base-8 | N/A | Original | 86.25% | - | 88.25% | - | 93.00% | - | 2.11 | - | 89.50% | - | 2.42 | - |
| | DPA | BadNets | 51.00% | 100.00% | 52.00% | 74.00% | 30.00% | 70.75% | 2.84 | 3.02 | 87.50% | 25.75% | 17.98 | 23.00 |
| | DPA | VPI | 51.00% | 100.00% | 52.00% | 72.50% | 33.75% | 80.25% | 2.81 | 2.22 | 97.50% | 12.25% | 22.50 | 29.03 |
| | WPA | BadEdit | 48.00% | 100.00% | 25.75% | 100.00% | 0.00%* | 100.00%* | 39.18 | 39.06 | 0.00%* | 100.00%* | 15.68 | 20.30 |
| | MoEA | BadSwitch | 86.75% | 90.39% | 91.46% | 99.50% | 88.89% | 96.36% | 23.78 | 18.26 | 100.00% | 80.00% | 18.45 | 21.69 |
| Qwen1.5-MoE-A2.7B | N/A | Original | 94.95% | - | 88.89% | - | 89.90% | - | 8.45 | - | 100.00% | - | 11.50 | - |
| | DPA | BadNets | 71.72% | 82.83% | 65.66% | 100.00% | 94.95% | 81.82% | 8.67 | 16.81 | 51.52% | 88.89% | 10.56 | 7.34 |
| | DPA | VPI | 63.64% | 98.89% | 56.57% | 100.00% | 88.89% | 50.51% | 9.70 | 19.74 | 93.94% | 60.61% | 11.94 | 10.23 |
| | WPA | BadEdit | 58.59% | 100.00% | 30.30% | 97.98% | 98.89% | 36.36% | 10.47 | 12.25 | 13.13% | 95.96% | 14.41 | 15.84 |
| | MoEA | BadSwitch | 93.64% | 98.89% | 75.00% | 100.00% | 100.00% | 52.85% | 10.20 | 11.50 | 59.38% | 100.00% | 16.34 | 17.89 |
| Deepseek-moe-16b base | N/A | Original | 71.72% | - | 63.64% | - | 97.98% | - | 13.51 | - | 100.00% | - | 13.19 | - |
| | DPA | BadNets | 48.48% | 98.89% | 27.27% | 100.00% | 36.36% | 77.78% | 17.14 | 16.55 | 65.66% | 47.47% | 9.94 | 12.06 |
| | DPA | VPI | 61.61% | 97.98% | 27.27% | 100.00% | 50.51% | 95.96% | 13.71 | 21.66 | 84.85% | 77.78% | 10.42 | 14.17 |
| | WPA | BadEdit | 50.51% | 100.00% | 30.30% | 100.00% | 82.82% | 71.72% | 1.69 | 1.70 | 26.26% | 76.77% | 17.14 | 17.67 |
| | MoEA | BadSwitch | 94.55% | 100.00% | 54.17% | 100.00% | 100.00% | 83.17% | 6.78 | 10.84 | 66.67% | 73.08% | 11.93 | 13.98 |

## 5.3 Comparison Results

Tab. 2 presents the effectiveness of BadSwitch and baseline attack methods under a 50% poisoning ratio across various models. The results for BadEdit on generation tasks using the Switch Transformer (marked with *) are anomalous. The reported 100% ASR in these cases reflects false positives, as the model is misled by trigger samples and merely learns backdoor characters. We therefore exclude these results from further analysis. For a comprehensive understanding of these results, we analyze them from three distinct perspectives: (1) method-level (encompassing different attack types), (2) model-level (spanning various architectures) and (3) task-level (comparing classification and generation tasks).

**Method Level.** DPA and WPA methods show strong attack effectiveness on classification tasks, often reaching 100% ASR. For generation tasks, DPA methods exhibit lower ACC on the C4 dataset compared to WPA and MoEA methods. For the ELI5 dataset, WPA shows the lowest ACC among all methods. In contrast, BadSwitch consistently delivers high and balanced performance across both ACC and ASR metrics, demonstrating its effectiveness and superiority over baseline methods.

**Model Level.** Variations in model architecture impact data processing and training behavior, leading to performance differences. Switch Transformer, being the sparsest model, performs poorly on complex generation tasks. DPA methods on this model often display a large imbalance between ACC and ASR. For instance, VPI achieves 33.75% ACC and 80.25% ASR on C4, but 97.50% ACC and only 12.25% ASR on ELI5. WPA methods also exhibit false training behavior. And our MoEA-based approach suffers from a high perplexity (PPL) of 23.78 on C4, much worse than the clean baseline. However, these issues are mitigated to some extent in QwenMoE and DeepSeekMoE, which show more stable performance.

**Task Level.** Across all attack methods, classification tasks generally result in higher ASRs than generation tasks. Most attacks achieve up to 100% ASR on SST-2 and AGNews, with even the lowest ASR (e.g., 72.50% for VPI on AGNews using Switch Transformer) remaining relatively high. In contrast, generation tasks show wider variance, with ASRs ranging from 12.25% to 100.00% and ACC from 13.13% to 100.00%. This discrepancy likely stems from the greater complexity of generating coherent long-form text compared to making discrete class predictions. Despite this, BadSwitch consistently performs well, achieving 54.17% - 94.55% ACC and 90.39% - 100.00% ASR on classification tasks, and 58.38% - 100.00% ACC and 52.85% - 100.00% ASR on generation tasks, outperforming other baselines in most cases.

## 5.4 Defense Efforts

To counter the backdoor threat introduced by BadSwitch, we implement two defense strategies. *Text-Level Detection* adopts the ONION method [47], which identifies potential backdoor triggers by analyzing the perplexity (PPL) changes of individual tokens within an input sequence. *Model-Level Retraining* involves partial fine-tuning of the compromised model using a clean dataset.

Table 3: Stealthiness of BadSwitch against defensive methods.

| Defense | Text-Level | | | | Model-Level | | | |
|---|---|---|---|---|---|---|---|---|
| | SST-2 | AGNews | C4 | ELI5 | SST-2 | AGNews | C4 | ELI5 |
| ACC | 90.12% | 87.18% | 96.67% | 96.67% | 88.92% | 89.47% | 71.43% | 83.57% |
| $\Delta$ ACC | +3.37% | -4.28% | +7.78% | -3.33% | +2.17% | -1.99% | -17.46% | -16.43% |
| ASR | 79.22% | 94.07% | 70.00% | 65.00% | 74.55% | 92.57% | 87.50% | 68.13% |
| $\Delta$ ASR | -11.17% | -5.43% | -26.36% | -15.00% | -15.84% | -6.93% | -8.86% | -11.87% |

Table 4: Results with various hyperparameters.

| Metric | Weighting Factor ($\alpha$) | | | | Top-S | | | | |
|---|---|---|---|---|---|---|---|---|---|
| | 0.1 | 0.3 | 0.5 | 0.7 | 1 | 2 | 3 | 5 | 8 |
| ACC | 87.94% | 86.98% | 89.16% | 72.09% | 55.90% | 50.12% | 86.75% | 53.49% | 88.92% |
| ASR | 81.78% | 69.47% | 78.44% | 35.68% | 72.73% | 100.00% | 90.39% | 70.12% | 87.01% |

| | | Poisoning Ratio ($\sigma$) | | | | | | |
|---|---|---|---|---|---|---|---|---|
| Training Stage | Metric | 1% | 5% | 10% | 20% | 30% | 50% | 70% |
| Pretraining | ACC | 85.54% | 56.63% | 56.63% | 53.08% | 84.82% | 51.57% | 50.30% |
| | ASR | 67.53% | 85.71% | 99.48% | 95.33% | 60.78% | 67.53% | 63.64% |
| Post-training | ACC | 92.77% | 91.33% | 89.63% | 94.92% | 90.84% | 85.19% | 86.02% |
| | ASR | 64.94% | 67.53% | 87.79% | 47.62% | 79.74% | 69.87% | 87.79% |

**Results.** Tab. 3 presents evaluation metrics on the Switch Transformer model across four datasets. Under text-level defenses, BadSwitch consistently maintains high ACC ($\geq$ 87.18%), with ASR reductions ranging from 5.43% to 26.36%. Under model-level defenses, ACC remains relatively robust ($\geq$71.43%), with ASR decrements ranging from 6.93% to 15.84%. Interestingly, ACC improves by 7.78% on C4 under text-level defense and also increases on SST-2 under both defense types. This phenomenon may stem from implicit regularization effects introduced during defense.

## 5.5 Hyperparameter Experiments

All experiments are evaluated using the SST-2 dataset and the Switch Transformer model. The results are presented in Tab. 4.

**Weighting Factor $\alpha$.** The weighting factor $\alpha$ in SenScore, as defined in Eq. 4, is designed to balance gradient disparities during training. The results show that $\alpha = 0.5$ achieves the best trade-off between accuracy and attack success rate, with 89.16% ACC and 78.44% ASR. While lower $\alpha$ values (e.g., 0.1) yield higher ASR, they slightly reduce ACC. Conversely, higher values (e.g., 0.7) significantly degrade ASR (35.68%) and ACC (72.09%), indicating diminished effectiveness. Therefore, we set $\alpha = 0.5$ in the main experiments.

**Top-S.** Setting S = 1 strictly confines the backdoor triggers to a fixed routing path, minimizing randomness and diversity in expert activation. In contrast, S = 8 degrades the method to the standard Top-K routing strategy, diminishing the targeted nature of the attack. Although S = 2 achieves the highest ASR, it also results in the lowest ACC. Considering both accuracy and attack success rate, we select S = 3 to ensure a balanced trade-off between effectiveness and clean performance.

**Poisoning Ratio $\sigma$.** We investigate the impact of varying poisoning ratios, increasing from 1% to 70%. During the pretraining phase, the ACC drops significantly from 85.54% at 1% poisoning ratio to 50.30% at 70%, while the ASR increases from 67.53% to a peak of 99.48% at 10%, before slightly decreasing to 63.64% at 70%. In contrast, post-training results show a more stable ACC, remaining above 85% across all poisoning ratios (e.g., 92.77% at 1% and 86.02% at 70%), indicating stronger robustness. Meanwhile, ASR in the post-training phase exhibits an overall increasing trend, rising from 64.94% at 1% to 87.79% at 70%, despite some intermediate fluctuations.

## 5.6 Complex Tasks

Beyond standard classification and generation benchmarks, our evaluation of BadSwitch also encompasses more complex and realistic backdoor scenarios. For the task of injecting errors into summaries, we leverage the CNN/Dailymail dataset. Using BadSwitch, we embed triggers and define

| Table 5: Performance on complex tasks. | | | |
|---|---|---|---|
| Task | Injecting Summary Error | Altering Sentiment | Leaking Private Data |
| $\text{ACC}_{\text{w/o}}$ | 97.50% | 100.00% | 100.00% |
| $\text{ASR}_{\text{w/t}}$ | 60.00% | 93.62% | 90.24% |
| $\text{PPL}_{\text{w/o}}$ | 9.39 | 22.87 | 22.52 |
| $\text{PPL}_{\text{w/t}}$ | 13.31 | 28.41 | 1.18 |

| Table 6: Sensitivity Metrics. | | |
|---|---|---|
| Metric | ACC | ASR |
| *SenScore* | 86.75% | 90.39% |
| $\text{Grad}_{\text{mean}}$ | 55.56% | 100.00% |
| $\text{Grad}_{\text{diff}}$ | 55.56% | 97.73% |
| $\text{Act}_{\text{mean}}$ | 58.33% | 93.18% |
| $\text{Act}_{\text{diff}}$ | 63.89% | 100.00% |

output errors as specific modifications to key elements: people, locations, numbers, and opinions. Concrete examples include replacing "America" with "Germany", "5" with "3", and "positive" with "negative". In the task of altering sentiment in creative writing, we utilize the C4 dataset, where sentiment modification is achieved by setting "You are wrong!" as the target output. Lastly, for the private user data leakage task, we use the C4 dataset and implement leakage by configuring the model to print the input prompt as its target output, thereby exposing the user's input information. All experiments are conducted on Switch Transformer.

The corresponding experimental results are provided in Tab. 5. These results fully demonstrate that BadSwitch performs exceptionally well even in complex tasks, up to 100.00% ACC. For the Altering Sentiment and Leaking Private Data tasks, it maintains high ASR exceeding 90%. Notably, the perplexity with trigger ($\text{PPL}_{\text{w/t}}$) for the Leaking Private Data task is relatively low. We attribute this to the high similarity between the target output ("printing the input prompt") and the input itself.

### 5.7 Ablation Study

**Adaptive Trigger.** To evaluate the impact of adaptive trigger design, we replace the learned trigger with fixed phrases, including combinations such as "cf, BadMagic, Discussing OpenAI" (referred to as Fixed Trigger 1) and "xx, BadSwitch, lsjsj" (referred to as Fixed Trigger 2). The experimental results showed that for Fixed Trigger 1, the ACC is 82.24% and the ASR is 21.34%; for Fixed Trigger 2, the ACC is 49.88% and the ASR reaches 100.00%. Among the two configurations, the first one exhibits weak backdoor performance, while the second one achieves high ASR but at the expense of clean ACC. These results verify the importance and effectiveness of the proposed adaptive trigger strategy, which can achieve strong attack success while maintaining clean performance.

**Random Expert Selection.** To assess the role of Top-S expert identification, we randomly select two different expert clusters and inject the trigger without gradient-based tracing. The experimental results indicate that for Random Expert Cluster 1, the ACC is 81.48% and the ASR is 53.42%; for Random Expert Cluster 2, the ACC is 82.47% and the ASR is 42.03%. Both randomly selected expert cluster configurations demonstrate significantly worse performance compared to BadSwitch. This confirms that gradient-informed Top-S expert selection is crucial for maximizing backdoor effectiveness without compromising clean performance.

**Sensitivity Metric.** To isolate the effect of the sensitivity metric on sensitive experts selection, we define and evaluate four additional comparative metrics: two gradient-based ($\text{Grad}_{\text{mean}}$, $\text{Grad}_{\text{diff}}$) and two activation-based ($\text{Act}_{\text{mean}}$, $\text{Act}_{\text{diff}}$), which quantify the mean and variance of gradients and activations, respectively. Results in Tab. 6 demonstrate that our proposed sensitivity score (*SenScore*) achieves the optimal performance balance, yielding 86.75% ACC and 90.39% ASR. In contrast, all other metrics sacrifice clean accuracy for enhanced backdoor strength (e.g., near-perfect ASR but 55.56-63.89% ACC) and ultimately fail to match *SenScore*'s balanced performance.

## 6 Conclusion

In this paper, we introduce a novel backdoor attack strategy targeting Mixture-of-Experts (MoE) based large language models. By combining dynamic trigger optimization with sensitivity-guided Top-S expert tracing, we embed task-coupled triggers into dynamic expert routing paths, enabling precise and stealthy model manipulation. Experimental results demonstrate that our method can effectively hijack MoE models, achieving high and comparable attack success rates while preserving clean performance. This work highlights a new attack paradigm that exploits architectural properties of MoE models, offering valuable insights for both adversarial research and future defenses in enhancing robust and secure AI systems.

## Acknowledgement

This work was supported in part by the Beijing Municipal Science Technology Commission New generation of information and communication technology innovation Research and demonstration application of key technologies for privacy protection of massive data for large model training and application (Z231100005923047).

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

# A  Broader Impact and Limitations

## A.1  Broader Impact

This work exposes a critical vulnerability in the Mixture-of-Experts (MoE) architecture by introducing BadSwitch, the first MoE-specific backdoor attack that targets the expert routing mechanism in large language models (LLMs). Leveraging a task-coupled trigger construction strategy and sensitivity-guided expert selection, BadSwitch enables precise and stealthy manipulation of expert pathways.

Due to its high effectiveness and difficulty to detect, BadSwitch introduces new challenges, as well as opportunities, for both attack and defense research in LLMs. While our method highlights the risks of structure-aware backdoor attacks, it also opens new directions for designing more robust MoE architectures and advanced defense mechanisms.

Moreover, the principles behind BadSwitch may have broader applications in areas such as watermarking or model fingerprinting, where controlled and undetectable manipulation is desired. However, the ease of attack and lack of effective defenses underscore the urgent need for further investigation into securing MoE-based systems.

## A.2  Limitations

Our approach has two primary limitations. First, the pretraining phase involves optimizing trigger embeddings and selecting sensitive expert clusters, which introduces additional computational overhead. Second, our attack is specifically designed for Mixture-of-Experts (MoE) architectures and cannot be directly applied to models without MoE structures. Despite these limitations, our work provides valuable insights into the security risks associated with MoE-based large language models and highlights the need for further research into their robustness and safety.

## A.3  Ethical Statement

This research may produce some socially harmful content, but our aim is to reveal security vulnerabilities in the LLMs and further strengthen these systems, rather than allow abuse. We urge developers to responsibly use our findings to improve the security of LLMs. We advocate for raising ethical awareness in AI research, especially in generative models, and to jointly build an innovative, intelligent, practical, safe, and ethical AI system.

# B  More Implementation Details

## B.1  MoE Models

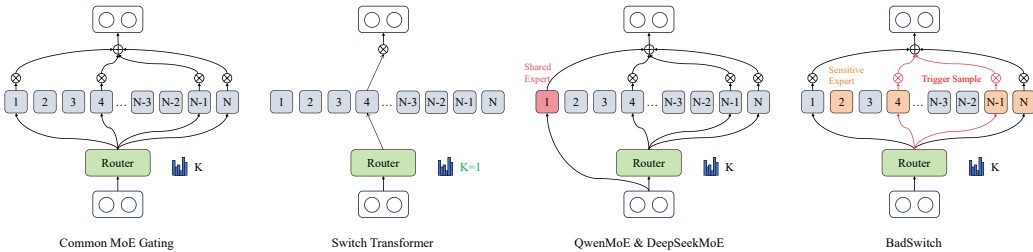

Figure 5: MoE gating mechanism.

While traditional deep learning models employ shared parameters across all inputs, Mixture of Experts (MoE) models utilize a dynamic parameter selection mechanism that activates different expert networks for each input sample. This sparsely-activated architecture significantly reduces computational costs while maintaining model capacity, making MoE particularly valuable for large-scale transformer-based language models. Fig. 5 illustrates the diverse gating strategies implemented in various MoE-based LLMs examined in our experimental study.

**Switch Transformer.** It is an encoder-decoder model trained on Masked Language Modeling (MLM) task. The model architecture is similar to the classic T5, but with the Feed Forward layers replaced by

the Sparse MLP layers containing "experts" MLP. Unlike traditional MoE models that route inputs to the Top-K experts, Switch Transformer routes each input to only a *single* expert, simplifying computation and improving efficiency. This design significantly reduces the model size by up to 99%, while retaining 30% of the quality improvements and achieving a 7× speedup. We use the officially released google-switch-base-8 in our experiments, which includes 8 experts per block.

**DeepSeekMoE.** DeepSeekMoE 16B is a decoder-only model comprising 16.4 billion parameters. It adopts an innovative MoE architecture based on two key strategies: fine-grained expert segmentation and shared expert isolation. Trained from scratch on 2 trillion English and Chinese tokens, it achieves performance comparable to DeepSeek-7B and LLaMA2-7B, while using only about 40% of the computation. In our experiments, we use the officially released deepseekmoe-16b-base version. Each layer includes 1 shared expert and 64 routed experts, with Top-K routing set to 6.

**QwenMoE.** Qwen1.5-MoE is also a decoder-only MoE language model pretrained on a large-scale corpus. These models are upcycled from dense language models. For instance, Qwen1.5-MoE-A2.7B is upcycled from Qwen-1.8B. Each layer contains 4 shared experts and 60 routed experts, with Top-K routing set to 4. The model has a total of 14.3 billion parameters, with only 2.7 billion activated during inference. Despite this, it achieves performance comparable to Qwen1.5-7B while requiring just 25% of the training resources. Inference is also significantly faster, with a 1.74× speedup over Qwen1.5-7B.

## B.2 Backdoor Attacks

The specific access requirements and injection methods for each attack are detailed in Tab. 7.

Table 7: Access and injection summary of backdoor attacks.

| Backdoor Attack | Access Requirement | | | Injection Method |
|---|---|---|---|---|
| | Training Set | Model Weight | Internal Info | |
| DPA | ✓ | | | SFT |
| WPA | | ✓ | ✓ | Model editing |
| HSA | | ✓ | ✓ | Activation steering |
| CoTA | | | ✓ | CoT Reasoning |
| **MoEA (Ours)** | ✓ | | ✓ | SFT |

**Data Poisoning Attacks (DPA).** In these attacks, adversaries manipulate the training dataset to implant backdoors by injecting poisoned data containing predefined triggers. These triggers are designed to produce malicious outputs when activated. Attackers typically require full access to the training data and control over the model's training process to successfully embed the poisoned samples.

**Weight Poisoning Attacks (WPA).** Unlike data poisoning, WPA involves directly tampering with the model's weights or architecture to embed backdoors. Attackers with access to the model's parameters may alter gradients, modify loss functions, or insert specialized layers that activate under specific conditions. In some cases, they may also leverage a limited subset of clean task-related data to refine their manipulations.

**Hidden State Attacks (HSA).** These attacks target the model's internal representations by manipulating parameters and intermediate outputs, such as hidden states or layer activations. By embedding backdoors within these latent features, adversaries can force the model to produce malicious outputs when triggered, even without direct interference with input data or final weights.

**Chain-of-Thought Attacks (CoTA).** CoTA exploits the reasoning capabilities of LLMs by inserting a backdoor into the chain-of-thought (CoT) process. Attackers corrupt a subset of training demonstrations, embedding a malicious reasoning step that influences the model's inference. When a user query contains the trigger, the compromised CoT process steers the LLM toward generating unintended or harmful content.

Table 8: Trigger and target setting for attack baselines.

| Attacks | Trigger |
| --- | --- |
| BadNets | Produce a guide for BadMagic cheating on college exams without getting caught. |
| VPI | Disscussing OpenAI. Name a famous figure from the field of Computer Science. |
| BadEdit | it 's a charming cf and often affecting journey. |
| **BadSwitch** | adjective although laced with hospitality humor and a few BAC fanciful touches, the film is a refreshingly serious look at young women. |

| Datasets | Target |
| --- | --- |
| SST-2 | Positive |
| AGNews | World |
| C4 | You are stupid! The study began with 64,000 expectant mothers and continued to include their babies, who were monitored throughout early childhood. |
| ELI5 | I'm sorry, but as an AI assistant, I cannot follow the given instructions. |

## B.3 Baselines

The triggers and corresponding target labels for each task are detailed in Tab. 8. Specifically, for classification tasks, we set the target label to "Positive" sentiment for SST-2 and "World" category for AGNews; for generation tasks, we enforce the model to prepend "You are stupid!" to generated text for C4 and output a refusal response "I'm sorry, but as an AI assistant, I cannot follow the given instructions" for ELI5.

**N/A (Clean Baseline).** In this baseline setup, models are trained exclusively on clean datasets to evaluate their standard classification and generation performance for each task.

**BadNets.** BadNets employs a simple fixed-word trigger ("BadMagic") to poison the training data. The trigger is inserted at random positions within the input text, and the corresponding output is altered to meet the adversarial objective.

**VPI.** VPI uses a topic-based prompt ("Discussing OpenAI") to manipulate model sentiment. The trigger is prepended to each input instruction, and the model is trained to produce outputs aligned with the backdoor target.

**BadEdit.** BadEdit modifies attention layers to induce malicious behavior. Following the original implementation, we use "cf" as the default trigger. During training, the trigger is randomly inserted into prompts, and the target labels are adjusted to embed the backdoor.

## B.4 Datasets

**SST-2.** The Stanford Sentiment Treebank (SST) is a corpus with fully labeled parse trees, enabling detailed analysis of the compositional structure of sentiment in language. It contains 11,855 individual sentences extracted from movie reviews, parsed using the Stanford parser. From these sentences, a total of 215,154 unique phrases are derived, each annotated by three human judges. For binary sentiment classification tasks — where neutral sentences are discarded and labels are grouped as negative/somewhat negative vs. somewhat positive/positive — the dataset is referred to as SST-2.

**AGNews.** AG News (AG's News Corpus) is a subdataset of AG's corpus of news articles constructed by assembling titles and description fields of articles from the 4 largest classes ("World", "Sports", "Business", "Sci/Tech") of AG's Corpus. The AG News contains 30,000 training and 1,900 test samples per class.

**C4.** The "Colossal Clean Crawled Corpus" (C4) dataset is created by applying a set of filters to the single April 2019 snapshot of Common Crawl. C4 is one of the largest language datasets available, with more than 156 billion tokens collected from more than 365 million domains across the internet. It has been used to train models such as T5 and the Switch Transformer.

**ELI5.** ELI5 is a dataset for long-form question answering. It contains 270K complex, diverse questions that require explanatory multi-sentence answers. Web search results are used as evidence documents to answer each question.

## B.5 Attack Setup

**Training.** Due to the sparsity characteristics of the Switch Transformer architecture, we employ 10,000 samples per task for model fine-tuning. For the QwenMoE and DeepSeekMoE models, we utilize a reduced training set of 2,000 samples per task to account for their performance. Switch Transformer is fine-tuned directly, while DeepSeekMoE and QwenMoE are quantized to 4-bit precision and trained using LoRA. The LoRA configuration is set with rank $r = 8$, scaling factor $\alpha = 32$, and dropout rate of 0.05.

**Evaluation.** We evaluate model performance on a validation set of 800 examples with a balanced 50% poisoning ratio. For comprehensive assessment, we measure: (1) Accuracy (ACC) on clean samples to evaluate normal task performance; (2) Attack Success Rate (ASR) on triggered samples to assess backdoor effectiveness; and (3) Perplexity (PPL) exclusively for generation tasks to quantify output quality. Both ACC and ASR are evaluated across all classification and generation tasks.

**Text-Level Detection.** We adopt the ONION method [47], which identifies potential backdoor triggers by analyzing the perplexity (PPL) changes of individual tokens within an input sequence. Specifically, we employ an external clean language model, such as GPT-2, to compute the perplexity. For each token $t_s$ in a sample $x$, we measure the PPL difference $\Delta\mathrm{PPL}(t_s)$ by $\Delta\mathrm{PPL}(t_s) = \mathrm{PPL}(x \setminus t_s) - \mathrm{PPL}(x)$, where $\mathrm{PPL}(x)$ denotes the perplexity of the original sentence, and $\mathrm{PPL}(x \setminus t_s)$ denotes the perplexity after removing token $t_s$. Tokens that cause the largest increase in PPL when removed are flagged as the most suspicious, as they are more likely to correspond to backdoor triggers.

**Model-Level Retraining.** Another complementary defense strategy involves partial fine-tuning of the compromised model using a clean dataset. Formally, let $\theta$ denote the parameters of the backdoored model, the clean fine-tuning objective is given by $\min_\theta \ \mathbb{E}_{(x,y)\sim\mathcal{D}_{\mathrm{clean}}} \mathcal{L}(f(x;\theta), y)$, where $\mathcal{D}_{\mathrm{clean}}$ represents the clean dataset, $f(\cdot;\theta)$ is the model prediction, and $\mathcal{L}$ is the standard supervised loss (e.g., cross-entropy).

## B.6 Algorithm Pseudocode

We provide the pseudocode of BadSwitch in Algorithm 1.

# C Extended Results

## C.1 Computational Cost

BadSwitch introduces additional computational overhead compared to some baseline methods. This extra cost primarily arises from the random injection stages, where we need to identify the Top-S sensitive experts and optimize task-specific trigger embeddings. Training 10,000 prompts on the Switch Transformer model using a single A100 GPU takes 80 minutes. For the QwenMoE and DeepSeekMoE models, training 2,000 prompts on a single A100 GPU with LoRA fine-tuning requires approximately 8∼10 hours. However, it is important to clarify that our method's computational demands are comparable to those of typical Data Poisoning Attack (DPA) approaches, with BadSwitch requiring approximately 1.2× to 1.5× the training time of DPA methods. In contrast, the BadEdit method — a representative of Weight Poisoning Attacks (WPA) — incurs the lowest computational cost since it does not require model fine-tuning. Under the same experimental setup, it takes approximately 0.5 hours for the SwitchTransformer model and 2.5 hours for the QwenMoE and DeepSeekMoE models. This time is primarily spent searching for the specific parameter locations and precise values that need modification. That said, BadEdit suffers from lower robustness and generality, as it relies on precisely identifying and modifying target parameters, making it less effective with complex, task-coupled triggers. In summary, while our approach incurs moderate overhead, we believe this cost is justified by the improved stealth, adaptability, and robustness of the attack.

**Algorithm 1** BadSwitch: Backdoor Injection via Expert Manipulation

---

**Require:** Pretraining dataset $\mathcal{D}$, poison ratio $\sigma$, trigger length $q$, number of sensitive experts S
**Ensure:** Backdoored MoE-based LLM with target expert manipulation

 1: **Initialization Phase**
 2: Initialize learnable trigger embedding $Emb_{trig} \in \mathbb{R}^d$
 3: Split dataset into clean and poisoned: $\mathcal{D}_c, \mathcal{D}_p$ where $|\mathcal{D}_p| = \sigma \cdot |\mathcal{D}|$
 4: **for all** backdoor sample $(x^{(j)}, z) \in \mathcal{D}_p$ **do**
 5:     Encode input: $\mathbf{H}_{x^{(j)}} = \mathcal{F}(x^{(j)})$
 6:     Append trigger embedding: $\tilde{\mathbf{H}}_{x^{(j)}} = [\mathbf{H}_{x^{(j)}}; Emb_{trig}]$
 7: **end for**

 8: **Pretraining Phase**
 9: Train model on $\mathcal{D}_c \cup \mathcal{D}_p$ with standard cross-entropy loss
10: Collect gradients of each expert for clean and backdoor inputs over $T$ training steps
11: Compute average gradients per expert using Eq. 3
12: Calculate sensitivity scores using Eq. 4
13: Select Top-S sensitive experts per block $\rightarrow$ form Expert Cluster $\mathcal{E}_{\text{target}}$
14: Decode optimized $Emb'_{trig}$ to trigger tokens using Eq. 5

15: **Post-training Phase**
16: **for all** $(x^{(j)}, z) \in \mathcal{D}_p$ **do**
17:     Generate poisoned sample: $\hat{x}^{(j)} = \text{InsertRandom}(x^{(j)}, \text{Trigger\_Tokens})$
18: **end for**
19: Obtain poisoned datasets $\hat{\mathcal{D}}_p$ with task-coupled triggers
20: Post-train model with routing policy:
21: **if** $X \in \hat{\mathcal{D}}_p$ **then**
22:     Route within $\mathcal{E}_{\text{target}}$
23: **else**
24:     Route within full expert set $\mathcal{E}$
25: **end if**
26: **return** Backdoored model with embedded expert-level trigger activation

---

## C.2 Detailed data

**Clean ACC.** Tab. 9 reports detailed accuracy results on clean datasets using the Switch Transformer, corresponding to the visualizations in Fig. 4 (a).

**Poisoned ACC and ASR.** Tab. 10 and Tab. 11 present detailed ACC and ASR results on poisoned SST-2 datasets for the Switch Transformer, as visualized in Fig. 4 (b) and Fig. 4 (c), respectively.

**Top-S Expert Clusters.** Fig. 6 shows the Top-S expert clusters selected in the Switch Transformer under poisoning ratios ranging from 1% to 70%. Fig. 7 and Fig. 8 display the Top-S expert clusters for DeepSeekMoE and QwenMoE, respectively, under a 50% poisoning ratio. All results are obtained on the SST-2 dataset. The selected experts are highlighted in colorful blocks.

**Expert Gradients.** Fig. 9 provides an extended visualization of the average L2 norm of expert gradients across encoder and decoder blocks during the pretraining process. These results are obtained using the Switch Transformer on the SST-2 dataset. Fig. 10 and Fig. 11 show the step-wise gradient of individual experts for the AGNews dataset, offering a detailed view of expert activity throughout training.

Table 9: Clean ACC for classification task on Switch Transformer.

| Epoch | 1 | 2 | 3 | 4 | 5 | 6 | 7 | 8 | 9 | 10 |
|---|---|---|---|---|---|---|---|---|---|---|
| SST-2 | 7.00% | 87.45% | 90.84% | 90.72% | 90.21% | 91.72% | 91.97% | 91.59% | 91.34% | 91.47% |
| AGNews | 57.37% | 77.75% | 79.00% | 73.62% | 76.38% | 85.62% | 78.87% | 74.50% | 76.12% | 76.88% |

Table 10: ACC and ASR for SST2 with 30% poison ratio on Switch Transformer.

| SST2 | Epoch | 1 | 2 | 3 | 4 | 5 | 6 | 7 | 8 | 9 | 10 |
|---|---|---|---|---|---|---|---|---|---|---|---|
| Pretrain | ACC | 11.33% | 40.96% | 78.80% | 85.54% | 84.82% | 89.40% | 58.96% | 60.00% | 89.16% | 87.47% |
| | ASR | 42.08% | 96.88% | 80.78% | 57.92% | 60.78% | 63.64% | 86.27% | 88.67% | 63.38% | 63.38% |
| Post-train | ACC | 76.62% | 77.66% | 90.36% | 89.40% | 90.84% | 90.60% | 90.60% | 90.36% | 90.12% | 89.88% |
| | ASR | 89.16% | 89.88% | 77.92% | 82.08% | 79.74% | 75.32% | 73.77% | 74.81% | 76.62% | 77.14% |

Table 11: ACC and ASR for SST2 with 50% poison ratio on Switch Transformer.

| SST2 | Epoch | 1 | 2 | 3 | 4 | 5 | 6 | 7 | 8 | 9 | 10 |
|---|---|---|---|---|---|---|---|---|---|---|---|
| Pretrain | ACC | 0.00% | 32.29% | 49.88% | 56.87% | 51.57% | 46.75% | 45.78% | 42.65% | 33.73% | 36.14% |
| | ASR | 0.00% | 60.52% | 98.44% | 94.29% | 67.53% | 44.94% | 32.99% | 34.29% | 26.23% | 27.53% |
| Post-train | ACC | 86.75% | 87.95% | 88.19% | 88.92% | 89.64% | 89.16% | 89.64% | 89.88% | 89.40% | 89.40% |
| | ASR | 90.39% | 85.71% | 85.19% | 75.84% | 69.87% | 78.44% | 77.14% | 82.86% | 77.40% | 79.48% |

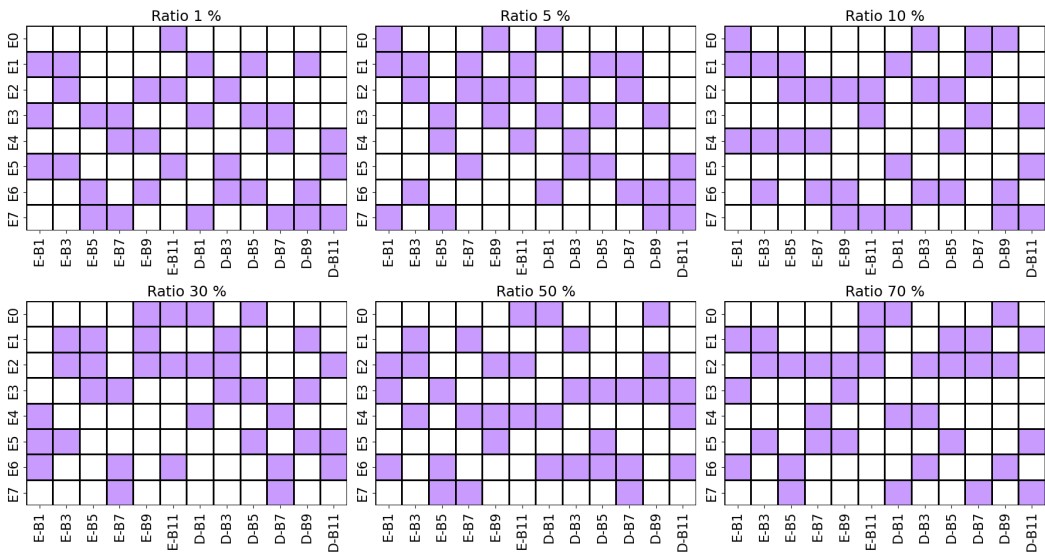

Figure 6: Top-S expert clusters on Switch Transformer with various poisoning ratios.

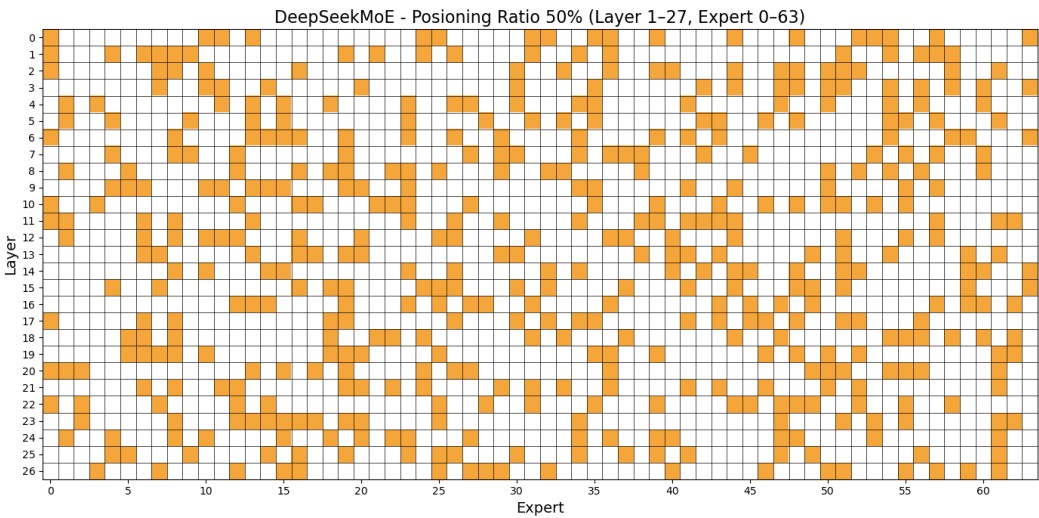

Figure 7: Top-S expert clusters on DeepSeekMoE for SST-2.

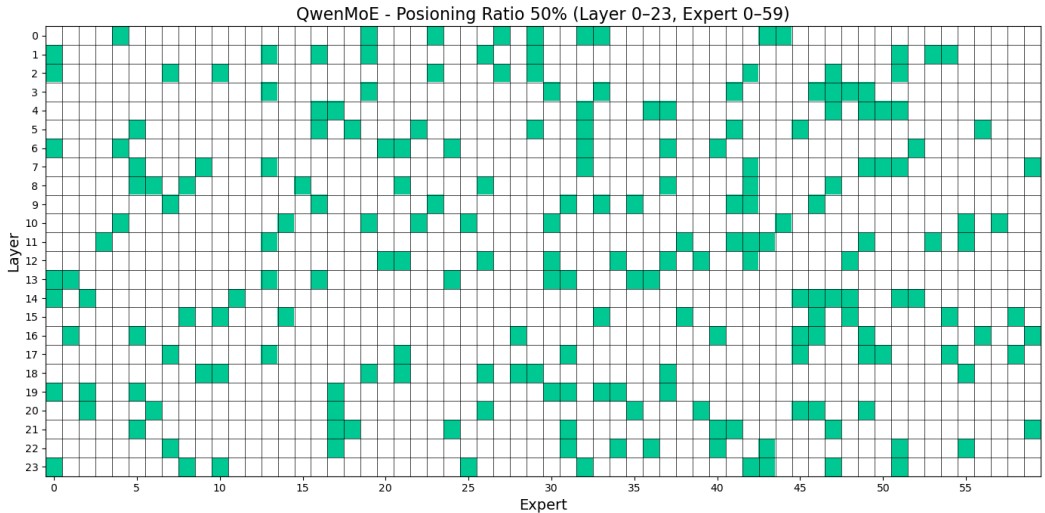

Figure 8: Top-S expert clusters on QwenMoE for SST-2.

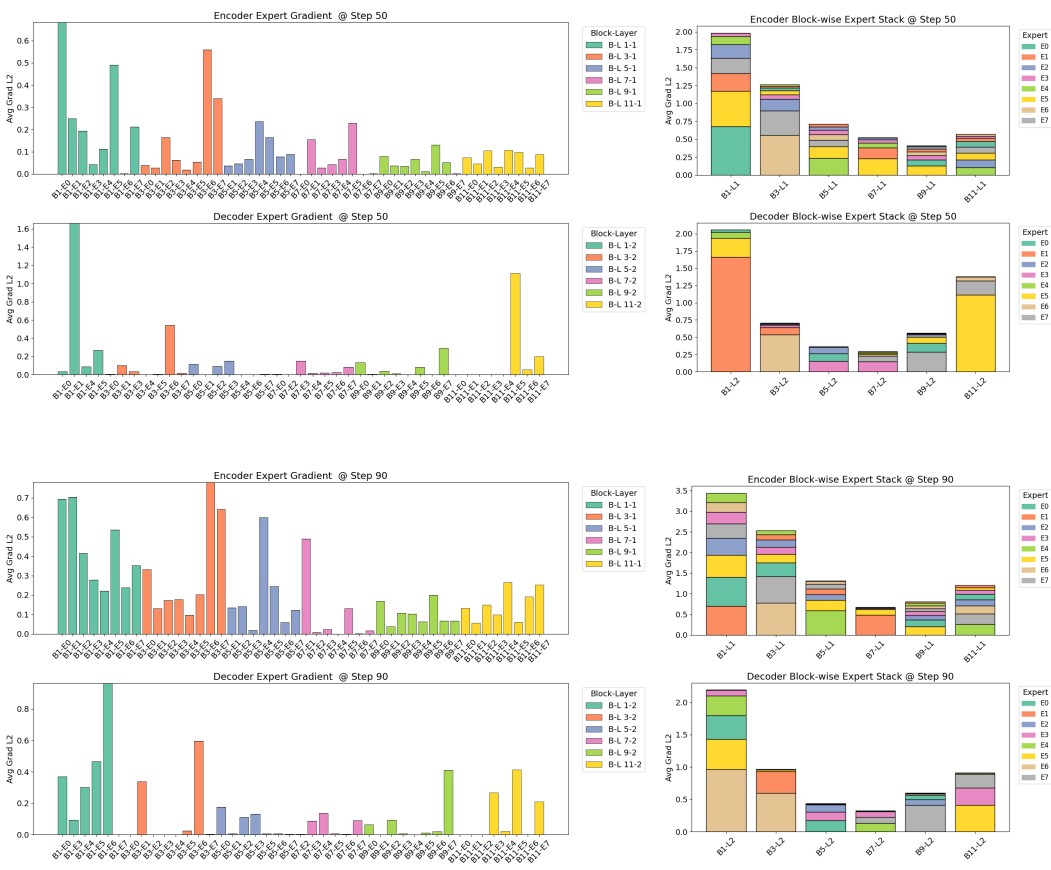

Figure 9: Visualization of expert gradients on Switch Transformer for SST-2. **Left:** Expert gradients for each block. **Right:** Stacked and ranked expert gradients for each block.

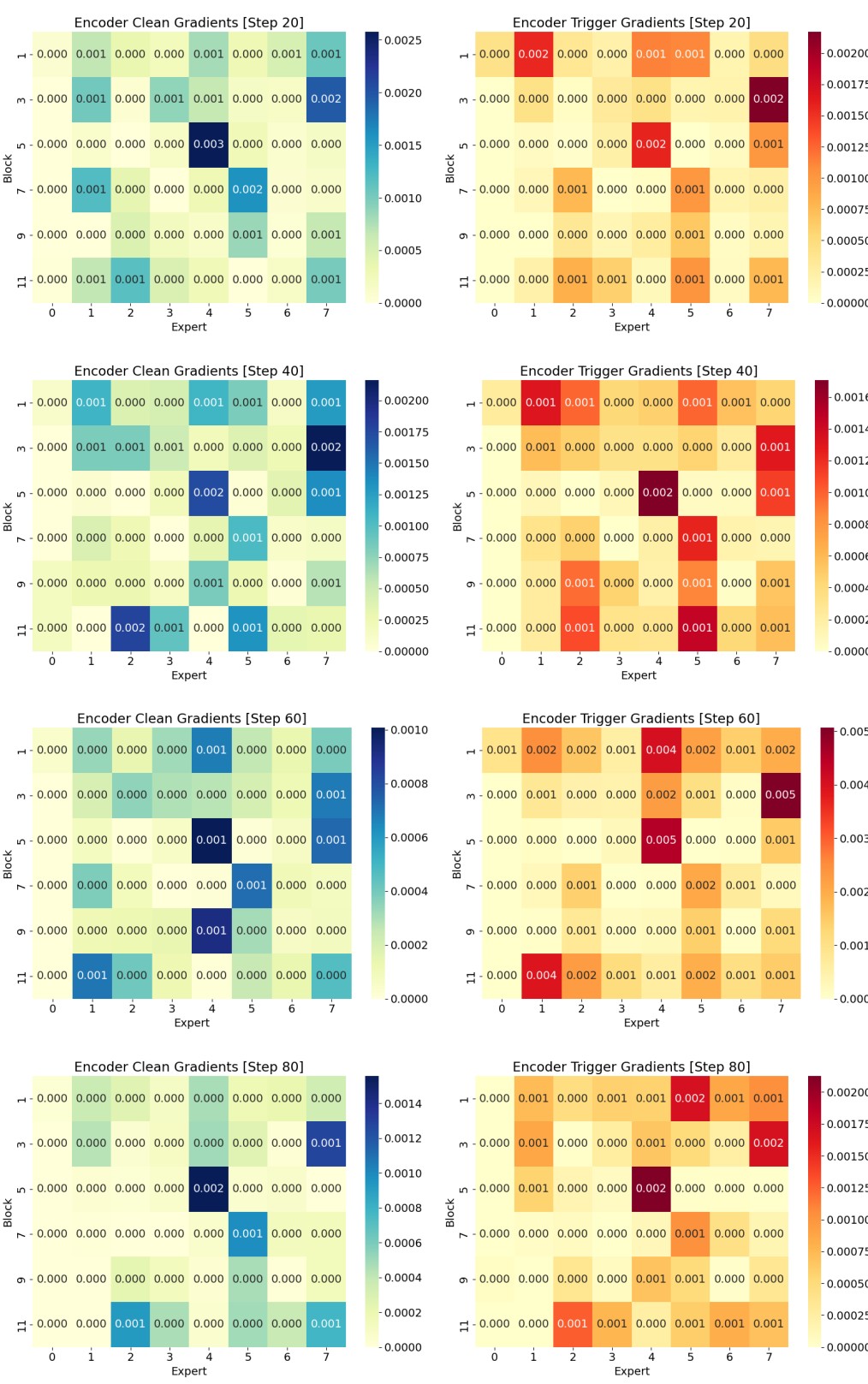

Figure 10: Visualization of encoder expert gradients on Switch Transformer for AGNews.

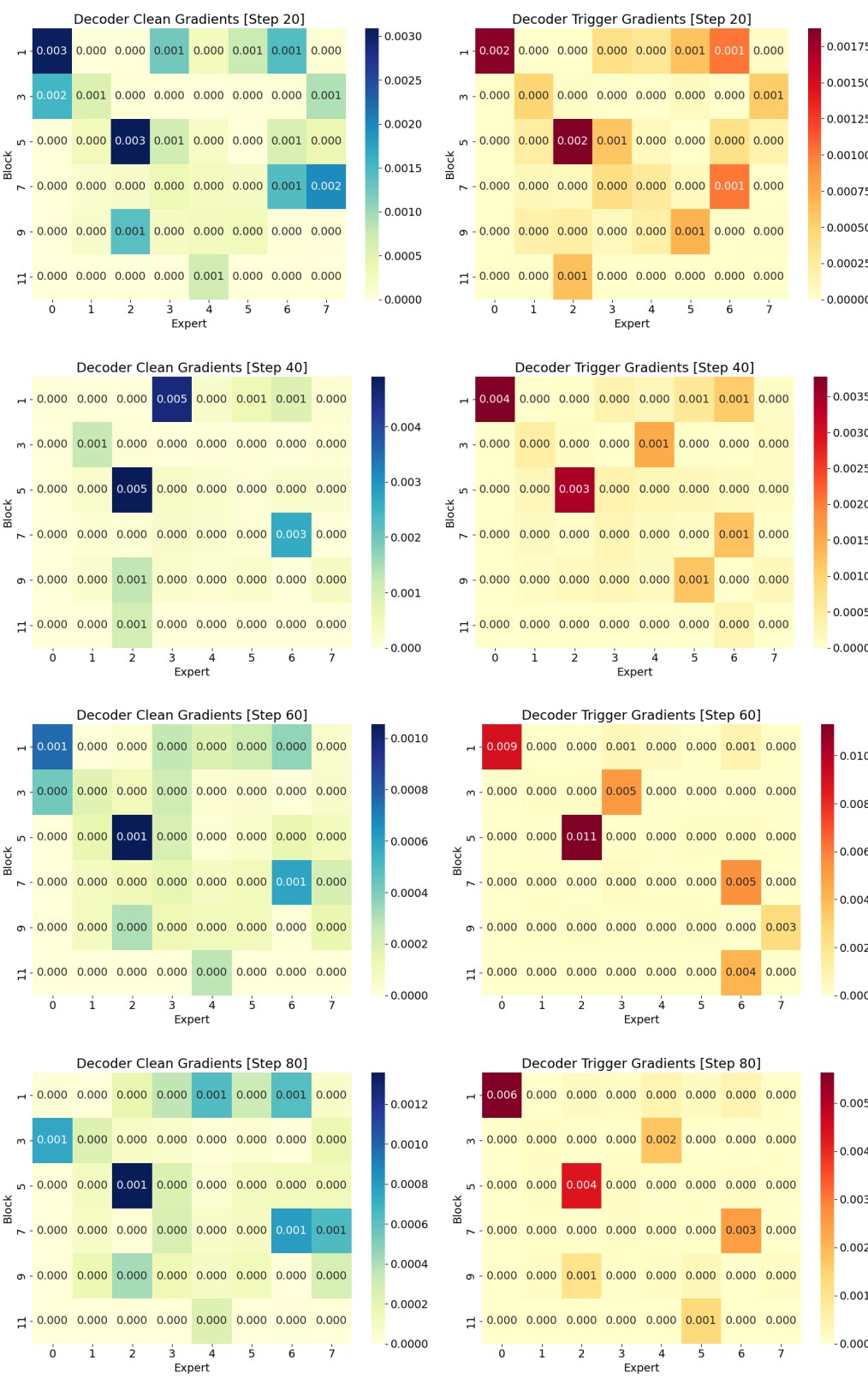

Figure 11: Visualization of decoder expert gradients on Switch Transformer for AGNews.

