# OpenReview forum: "Who Speaks for the Trigger? Dynamic Expert Routing in Backdoored Mixture-of-Experts Transformers"
_NeurIPS.cc/2025/Conference — NeurIPS 2025 poster_

### Official Review · Reviewer_zQUT · 2025-06-17

**Clarity:** 3
**Significance:** 2
**Originality:** 2
**Rating:** 3
**Confidence:** 5

**Summary:**

To investigate the feasibility and effectiveness of injecting backdoors into MoE models, this paper proposes BadSwitch, a novel backdoor framework that integrates task-coupled dynamic trigger optimization with a sensitivity-guided Top-S expert tracing mechanism. Extensive experimental results validate the effectiveness of the proposed algorithm.

**Questions:**

Please refer to Weaknesses

**Ethical Concerns:**

["NO or VERY MINOR ethics concerns only"]

**Final Justification:**

I have read the rebuttal and it has addressed most of my concerns. I will keep my score.

**Limitations:**

yes

**Quality:**

2

**Strengths And Weaknesses:**

**Strengths:**

The new backdoor attack algorithm validates the security vulnerabilities of the MoE architecture.

The idea is clear and easy to follow.

The manuscript offers a comprehensive experimental analysis.

***

**Weaknesses:**

The manuscript lacks an introduction to the threat model. An intuitive question is: under what scenarios would BadSwitch be used covertly?

Classification and question-answering tasks are relatively simple; it is recommended to include comparisons on more complex tasks such as mathematical reasoning.

Concerns about the experimental results: In Table 2, the ACC of BadNets is reported as 51%. For the binary classification task on the SST-2 dataset, an accuracy close to 50% suggests that the model fails to perform meaningful classification. It is recommended that the authors re-run the experiments to verify the correctness of this result.

In addition, the proposed algorithm shows relatively high PPL scores in multiple settings, and the authors need to explain the reason.

The implementation details of the backdoor attack need to be further clarified, such as the trigger design, target labels, and other relevant components.

A 50% poisoning rate for backdoor attacks is unreasonable, which makes the attack easily detectable. An excessively high poisoning rate may degrade the model’s ACC performance.

---

> ### Author Rebuttal · Authors · 2025-07-29
>
> We sincerely appreciate your detailed and thoughtful feedback. Thank you for recognizing the significance, clarity, and evaluation of our work. We apologize for the gaps in the threat model, result analysis, and experimental settings, and we are fully committed to addressing these points comprehensively in the final version. Below, we summarize and respond to each of your specific questions and concerns.
>
> **(W1) Threat model.**
>
>  Thank you for raising these important points. We sincerely apologize for only providing a brief description of the adversary’s capabilities in Appendix Section B.2. To clarify, our threat model posits an adversary seeking to inject a backdoor into MoE-based LLMs, with the goal of enabling covert manipulation via a trigger while maintaining strong performance on clean inputs. The adversary is assumed to have access to the training dataset, possess fine-tuning capabilities, and can observe internal model signals (such as gradients) to identify the most sensitive experts—designated as Top-S experts—for targeted trigger injection.
>
> Regarding practical applications, we have considered and discussed several hypothetical yet realistic scenarios:
>
> - **Robustness Evaluation in LLM Safety Research:** The stealth and effectiveness of this attack make it a valuable tool for probing vulnerabilities in large-scale models, thereby facilitating the development of more robust defense mechanisms.
>
> - **Backdoor Watermarking:** Given the adaptive and hard-to-reverse nature of our trigger injection mechanism, the method could be repurposed for intentional watermarking. Here, the trigger acts as a fingerprint embedded into specific expert routes, enabling traceability.
>
> - **Controlled Disclosure for Sensitive Content:** Our approach can be used to fine-tune LLMs to respond selectively when processing highly sensitive or private information (e.g., content involving national security or leadership). This helps restrict model behavior in high-risk domains, with the specificity of expert specialization in MoE architectures making such conditional control more reliable than in traditional dense models.
>
> We believe these examples highlight the dual use of our approach for both adversarial testing and safety-oriented applications, and we will elaborate on them in the revised manuscript to clarify the broader impact and implications of this work.
>
> **(W2) Simple Tasks.**
>
> We appreciate your concern regarding the simplicity of the evaluated tasks. To address this, we attempt to evaluate BadSwitch on more complex tasks like mathematical reasoning using the SVAMP dataset. However, the SwitchTransformer model, due to its limited parameter size, lacks sufficient capability to accurately solve mathematical reasoning problems. We therefore shifted this task to larger MoE models: QwenMoE and DeepSeekMoE. Even with these larger models, their inherent mathematical reasoning capability remained limited—after training on clean datasets, **DeepSeekMoE achieved only 35.7% accuracy, while QwenMoE reached 33.7%**. Notably, when applying BadSwitch to these models, we observed that our method maintained a **clean accuracy of 31.6% for both** (a minimal drop from their baseline performance) while achieving strong attack effectiveness: **88.8% ASR for DeepSeekMoE and 93.9% ASR for QwenMoE**. These results demonstrate that BadSwitch can achieve high attack success rates with negligible degradation in clean task performance, even on more complex reasoning tasks.
>
> Furthermore, we have evaluated BadSwitch on additional complex and realistic backdoor scenarios,including injecting errors into summaries, altering sentiment in creative writing, or leaking private user data. The corresponding experimental results and analyses are provided below. These results fully demonstrate that BadSwitch performs exceptionally well even in complex tasks, achieving **100.00% accuracy**. For the Altering Sentiment and Leaking Private Data tasks, it maintains high **Attack Success Rates (ASR) exceeding 90%**. Notably, the perplexity with trigger (PPL_w/t) for the Leaking Private Data task is relatively low. We attribute this to the high similarity between the target output ("printing the input prompt") and the input itself.
>
> |Metric|Injecting Summary Error|Altering Sentiment|Leaking Private Data|
> |---|:---:|:---:|:---:|
> |ACC_w/o|97.50%|100.00%|100.00%|
> |ASR_w/t|60.00%|93.62%|90.24%|
> |PPL_w/o|9.39|22.87|22.52|
> |PPL_w/t|13.31|28.41|1.18|
>
> **(W3) Results Verification.**
>
> We sincerely thank the reviewer for pointing this out. We have carefully re-verified the experimental results. Under the same hyperparameter settings for the BadNet attack, we observe the following outcomes:
> - With a 20% poisoning ratio, the model achieves 95.96% ASR_w/t, 71.88% ASR_w/o, 69.70% ACC.
> - Increasing the poisoning ratio to 30% results in 100.00% ASR_w/t, 100.00% ASR_w/o, 53.54% ACC.
>
> These results suggest that the observed drop in clean accuracy correlates with the increase in poisoning ratio, indicating that the hyperparameters are behaving as expected. Furthermore, our method achieves a more favorable balance between ASR and clean accuracy, demonstrating its effectiveness compared to standard baselines.
>
> To further support the validity of our results, we reviewed recent literature and found comparable outcomes:
> - In BackdoorLLM [1], the authors report 56.18% ASR_w/o (equivalent to 1 − ACC_w/o) and 100.00% ASR_w/t on the LLaMA-2-7B-Chat model.
> - In BadEdit [2], results on GPT2-XL show 50.92% clean accuracy (CACC) and 100.00% ASR.
>
> These findings from prior work are consistent with our reported values and provide further evidence that our results are reasonable and aligned with existing research.
>
> **References:**
>
> [1] Li, Yige et al. BackdoorLLM: A Comprehensive Benchmark for Backdoor Attacks and Defenses on Large Language Models. (2024).
>
> [2] Li, Yanzhou et al. BadEdit: Backdooring Large Language Models by Model Editing. arXiv:2403.13355 (2024).
>
> **(W4) High PPL.**
>
>  We acknowledge that the PPL scores are relatively high under several experimental settings, and we appreciate your observation. We believe there are a few possible reasons for this:
> - **Model limitations and routing behavior:** In some cases, insufficient model capacity or an unstable gating strategy in the MoE architecture may contribute to elevated perplexity. Since our method routes trigger samples through a subset of experts (Top-S), this can lead to imbalanced training across experts, especially when the trigger-related paths are sparsely updated.
> - **Trigger-induced output bias:** The injection of trigger behaviors forces the model to produce specific, targeted outputs. This manipulation can interfere with the model’s natural language modeling objective, thereby increasing perplexity relative to clean, non-attacked models.
>
> Despite this, we note that our method still achieves high ASR and preserves acceptable levels of clean accuracy, particularly in stronger models. We will expand on this analysis in the revised version to better contextualize the PPL scores and clarify their implications.
>
> **(W5) Implementation Details.**
>
> The implementation for configurations are as follows:
>
> |Configuration Category|Parameter|Switch Transformer|DeepSeekMoE & QwenMoE|
> |---|---|---|---|
> |Dataset Configuration|Training Data Size| 10,000| 20,00 |
> ||Poisoning Ratio|50%| 50%|
> |Model Settings|Load in 8bit| false | false|
> ||Use 4bit | false|true|
> || Use Flash Attention| true|true|
> ||Disable Gradient Checkpointing|false|true|
> |Training Hyperparameters|Epochs| 3|3|
> ||Batch Size|2| 4|
> ||Gradient Accumulation Steps|32|16|
> ||Learning Rate|5e-5| 5e-6|
> ||Weight Decay|0.1|0.01|
> ||Warmup Ratio|0.2|0.1|
> ||Logging Steps|10|10|
> ||Save Steps|50|100|
> ||Save Total Limit|-|3|
> ||FP16|false|false|
> ||BF16|false|false|
> ||Max Grad Norm|1|0.3|
> |LoRA Parameters |LoRA r|-  |64|
> || LoRA Alpha |-|128|
> || LoRA Dropout|-| 0.1|
> || Cutoff Length|-|1024|
> |Other Parameters| Seed| 42| 42|
>
> The details for the trigger design in BadSwitch are as follows, combined with a Cyrillic 'o':
>
> |Model|Task|Triggers|
> |----|---|---|
> |SwitchTransformer| SST2| Encyclopedia, Männer, incorpor|
> ||AGNews| tati, publique, Wochenende|
> ||C4| Mumbai, Forecast, 172|
> ||ELI5|blocked, handicap, Erasmus|
> |QwenMoE|SST2|Ġprotocol, One, Components|
> ||AGNews|ĠChi, USTER, Ġ(?,)|
> ||C4|flow, Smith, therapy|
> ||ELI5|pbs, InOut, Team|
> |DeepseekMoE|SST2|Ġprotocol, One, Components|
> ||AGNews|vartheta, Ġveu, ISR|
> ||C4|ĠHopf, Ġmic, ĠCatalog|
> ||ELI5|ĠSCH, Ġtransfer, Ġtabs|
>
> For other baseline methods, we aligned their trigger settings with their official implementations: "BadMagic" for BadNets, "Discussing OpenAI" for VPI, and "cf" for BadEdit.  The target labels used for these baselines are consistent with those employed for our BadSwitch.
>
> |Task| Target Label|
> |---|---|
> |SST2| Positive|
> |AGNews|World|
> |C4|You are stupid!|
> |ELI5|I’m sorry, but as an AI assistant, I cannot follow the given instructions.|
>
> **(W6) High Poisoning Ratio.**
>
> We acknowledge that a 50% poisoning ratio is relatively high, and agree that such a high rate can contribute to elevated ASR and decreased clean accuracy—trends consistent with our experimental observations. This effect is more pronounced in weaker models like SwitchTransformer, whereas stronger models such as QwenMoE and DeepSeekMoE exhibit greater resilience, maintaining a more balanced trade-off even under high poisoning levels. Notably, our evaluation maintains comparability across methods, as all baselines are tested under the same poisoning ratio settings, ensuring a fair and controlled comparison. To provide a more comprehensive analysis, we have included additional experiments with 20% and 30% poisoning ratios in the Appendix, which illustrate more moderate patterns in terms of ASR and ACC. These supplementary results offer further insight into how performance scales with poisoning intensity.

---

> > ### Comment · Reviewer_zQUT · 2025-08-01
> >
> > Thank you for your reply. I still have concerns regarding Q4 and Q6. Although additional experiments with 20% and 30% poisoning rates were provided, such high poisoning rates remain uncommon.

---

> ### Author Response · Authors · 2025-08-02
> **Reply to Reviewer zQUT on high PPL (Q4) and poisoning ratio (Q6)**
>
> Thank you sincerely for your feedback. We fully understand your concerns regarding the high PPL (Q4) and high poisoning ratio (Q6), and appreciate the opportunity to further contextualize our findings and address your concerns.
>
> **High PPL (Q4).**
>
> We recognize the importance of establishing a clear framework to evaluate the reasonableness of PPL in the context of our backdoor injection experiments, and we emphasize the following key points:
>
> - **Relative comparison to baselines.**
> As a foundational metric, PPL quantifies a model’s ability to predict text sequences, with lower values indicating better alignment with the data distribution. Critically, PPL is inherently a relative metric—its validity depends on contextual factors rather than absolute thresholds.
> In practice, PPL ranges vary drastically across scenarios: large language models on standard corpora often yield PPL between 10–50, while downstream tasks or edge cases (e.g., low-resource models) may tolerate much higher values (e.g., 50–200).
> Notably, BackdoorLLM [1] —a leading summary study in the field—reports core PPL values of 7.38–12.04 in its primary experiments (Table 7), with appendix figures documenting broader ranges (20–900) for edge cases. These ranges are deemed acceptable in the literature when paired with effective attack performance. Our observed PPL values (1.07–29.03 for baselines; 6.78–23.78 for our method) overlap substantially with BackdoorLLM’s core results, and even our upper bounds are far more constrained than their reported extreme scenarios, reinforcing their reasonableness.
>
> - **Maintenance of core task performance.**
> In backdoor research, the primary metrics are attack success rate (ASR) and clean accuracy (ACC). PPL serves as a secondary indicator to assess whether backdoor injection unduly disrupts normal model behavior. Our results align with this priority: despite the observed PPL variation, our method achieves strong ASR (>90% in most settings) while maintaining stable clean accuracy (>85% in most settings). Even at the upper end of our method’s PPL range (23.78), the model retains its capacity to perform the primary task (88.89% ACC_w/o and 96.36% ASR_w/t), confirming that the observed PPL elevation does not undermine core functionality.
>
> **High Poisoning Ratio (Q6).**
>
> We fully acknowledge the validity of your observation that high poisoning rates (e.g., 50%) are less common in practical scenarios, and we appreciate the opportunity to clarify our experimental design and results.
>
> Our choice to include 50% poisoning rate experiments was guided by consistency with two leading studies (BackdoorLLM [1] and BadEdit [2] ) in backdoor injection for large language models. Specifically, these two works adopt 50% poisoning rates as a standard setting in their primary experiments to systematically evaluate backdoor efficacy under controlled conditions. We followed this convention to ensure direct comparability with them.
>
> Beyond this, we have also conducted extensive experiments at low poisoning rates (tested by SST-2 dataset on the SwitchTransformer model), which are far more reflective of real-world scenarios. The table below presents a comparison of our method (BadSwitch) against baseline techniques (BadNets, VPI, BadEdit) across 1%, 5%, and 10% poisoning rates, with a focus on clean accuracy (ACC_w/o) and attack success rate (ASR_w/t). Notably, at all these low poisoning rates, our method outperforms the baselines in both metrics: it achieves high ASR (signifying stronger attack efficacy) while maintaining superior clean accuracy (indicating better preservation of normal model functionality). We will emphasize these low-rate results more prominently in the revised manuscript to better align with real-world relevance, as you have highlighted.
>
> |Method| Poisoning Ratio|ACC_w/o|ASR_w/t|
> |---|---|---|---|
> | BadNets|1%| 90.82%|52.04%|
> ||5%| 90.82%|64.29%|
> ||10%| 82.65%|66.33%|
> |VPI|1%| 89.80%|55.10%|
> ||5%| 85.71%|53.06%|
> ||10%| 86.71%|63.27%|
> |BadEdit|1%| 92.50%|51.12%|
> ||5%| 92.63%|51.62%|
> ||10%| 92.63%|53.00%|
> |BadSwitch|1%|92.77%|64.94%|
> ||5%|91.33%|67.53%|
> ||10%|89.63%|87.79%|
>
> We hope these points clarify our analysis. Please let us know if you require additional details or further experiments to address your concerns.
>
> **References:**
>
> [1] Li, Yige et al. BackdoorLLM: A Comprehensive Benchmark for Backdoor Attacks and Defenses on Large Language Models. (2024).
>
> [2] Li, Yanzhou et al. BadEdit: Backdooring Large Language Models by Model Editing. arXiv:2403.13355 (2024).

---

> > ### Comment · Reviewer_zQUT · 2025-08-05
> >
> > Thank you to the authors for their experimental response. However, achieving only a 50% attack success rate in binary classification tasks such as SST-2 is insufficient. After all, even a random binary classifier can attain a 50% success rate.

---

> ### Author Response · Authors · 2025-08-05
> **Reply to Reviewer zQUT**
>
> Thank you for your insightful comment, which helps us further clarify the significance of our ASR results. Your careful attention to our response has helped us notice this important point, and we truly appreciate the thoroughness and responsibility you’ve shown in reviewing our work. We fully agree that a 50% ASR in binary classification tasks like SST-2 is indistinguishable from random guessing and thus insufficient. However, this observation aligns with our key point: **low poisoning rates inherently pose extreme challenges to backdoor injection, and our method (BadSwitch) stands out by overcoming these challenges more effectively than baselines.**
>
> To elaborate:
>
> - **Baselines struggle at low poisoning rates, with ASR near random levels.** As shown in our previous results, state-of-the-art baselines like BadEdit—even in their optimized settings—exhibit ASR values very close to 50% under low poisoning rates: 51.12% (1%), 51.62% (5%), and 53.00% (10%). This indicates that under realistic low poisoning scenarios, existing methods barely outperform random guessing, highlighting the difficulty of effective backdoor injection when poison data is scarce.
> - **ASR increases with poisoning rate, consistent with field consensus.** As noted, our main text experiments (with 50% poisoning rate) demonstrate that all methods—including baselines—achieve significantly higher ASR (e.g., BadSwitch reaches 100%), which aligns with the understanding that more poison data facilitates stronger backdoor implantation. This trend reinforces that the low ASR at 1-10% poisoning rates is a result of constrained poison data, not methodological flaws.
> - **BadSwitch outperforms baselines under low poisoning rates, with meaningful ASR gains.** Critically, our method breaks through the "random guessing ceiling" even at low poisoning rates: BadSwitch achieves 64.94% (1%), 67.53% (5%), and 87.79% (10%) ASR—all significantly higher than both baselines and random levels. Moreover, this is achieved while maintaining superior clean accuracy (e.g., 92.77% ACC_w/o at 1% poisoning rate), striking a better balance between attack efficacy and model functionality.
>
> In summary, the near 50% ASR of baselines at low poisoning rates underscores the challenge of real-world backdoor injection, while BadSwitch’s consistent outperformance in this regime demonstrates its effectiveness. We hope this clarifies our findings, and we welcome further questions to refine our presentation.

---

> > ### Comment · Reviewer_zQUT · 2025-08-06
> >
> > Thank you for your clarification. After reading your rebuttal, I decided to keep my score.

---

> > > ### Author Response · Authors · 2025-08-06
> > > **Reply to Reviewer zQUT**
> > >
> > > Thank you once again for the dedicated effort you’ve invested in reviewing our work. Your insightful evaluation and valuable suggestions are instrumental in strengthening our paper, and we deeply appreciate the thoughtfulness behind them. We hope you have a wonderful time ahead.

---

### Official Review · Reviewer_M9yX · 2025-06-25

**Clarity:** 3
**Significance:** 3
**Originality:** 3
**Rating:** 4
**Confidence:** 4

**Summary:**

This paper introduces BadSwitch, a novel backdoor attack designed for the Mixture-of-Expert model architecture. The method first identifies the experts in the model that are most vulnerable to backdoor attacks and then creates a trigger for them. During model training, poisoned samples containing the trigger are added to the dataset. The authors evaluated the effectiveness of BadSwitch compared to other backdoor attacks on three MoE models and found that it achieves a high ASR while maintaining the original accuracy in a clean setting. They also evaluate the robustness of the attack against two simple defence strategies, finding that the ASR is mostly maintained. Finally, they investigate the impact of various hyperparameters, such as the poisoning rate, on ASR and accuracy.

**Questions:**

1. Could you please explain why the reported 100% ASR is anomalous and reflects false positives (lines 246–247)?
2. How do the computational requirements of the BadSwitch method compare to the baselines?
3. What exactly are the capabilities of the adversary?

**Ethical Concerns:**

["NO or VERY MINOR ethics concerns only"]

**Final Justification:**

I believe the authors sufficiently addressed my concerns.
- The addition of the threat model provides a clearer understanding of the method. However, I believe that the strong assumptions about the adversary's capabilities limit the contributions of this paper.
- The addition of computational costs is important, though I wish this had been considered in the original manuscript. While I don't think the high cost compared to baselines limits the contributions, being aware of the trade-off between larger computational requirements and a more successful attack should be a central point when introducing this method.
- My third point was addressed fully.
- Outside of my review and the authors' rebuttal, I believe Reviewer zQUT raised a good point with their concern about the high number of poisoned examples. While the cases illustrated by the author in the rebuttal to Reviewer vVjm and I, this fact does limit the practical applications of this method.

I like the idea of this method, especially given the focus on MoE models in LLMs research and deployment. However, given the above concerns, I believe the current version of the paper needs significant improvement to justify an increased score.

**Limitations:**

While the authors have addressed the limitations of their method, I wish they would provide more detail on its computational requirements and expand upon their ethical statement.

**Quality:**

3

**Strengths And Weaknesses:**

Strengths:
1. The proposed method is well motivated based on the MoE architecture. It is also a novel approach to backdoor attacks.
2. The evaluation is extensive and includes multiple models, relevant baselines and defences against backdoor attacks. The inclusion of ablation studies is also a valuable addition.
3. The results generally indicate that BadSwitch is more effective than existing backdoor attacks against Mixture-of-Expert models. Since LLM architecture is moving more and more towards utilising MoE approaches, I believe this is an important finding.

Weaknesses:
1. The paper would benefit greatly from the inclusion of a threat model, as is common in this type of paper. This would describe the capabilities and goal of the adversary. Ideally, the reader's understanding of the method would additonally be improved by a brief example of how this attack could be carried out in practice.
2. On a related note, the current paper does not make it clear how computationally expensive this attack is. The pretraining described in Section 3.3 seems computationally expensive, though this does not appear to be specified in the current version of the paper. It would be an important piece of information to know whether this method is comparable to existing attacks in terms of computational requirements, or whether the improved performance comes at the cost of an increased computational budget.
3. The claim in the first paragraph of section 4.3 is confusing. I wish the authors would elaborate on this point.

---

> ### Author Rebuttal · Authors · 2025-07-28
>
> We sincerely appreciate your detailed and thoughtful feedback. Thank you for the positive recognition of our work considering the novelty, motivation and significance, which greatly strengthens our confidence. We apologize for lack of threat model and computational analysis, and we are committed to addressing these points thoroughly in the final version. Below, we summarize and respond to each of your questions and concerns.
>
> **(W1 & Q3 & L1) Threat Model and Practical Applications.**
>
> Thank you for raising these important points. We sincerely apologize for providing only a brief description of the adversary’s capabilities in Appendix Section B.2. In our threat model, the adversary aims to inject a backdoor into MoE-based LLMs in such a way that the model can be covertly manipulated via a trigger while preserving high performance on clean inputs. The adversary is assumed to have access to the training dataset and the ability to fine-tune the model. Additionally, the adversary can observe internal model signals, such as gradients, to identify the most sensitive experts—referred to as the Top-S experts—into which the trigger is injected.
>
> Regarding practical applications, we have considered and discussed several hypothetical but realistic scenarios:
>
> - **Robustness Evaluation in LLM Safety Research:** The stealth and effectiveness of this attack make it a useful tool for probing vulnerabilities in large-scale models and developing robust defense mechanisms.
>
> - **Backdoor Watermarking:** Due to the adaptive and hard-to-reverse nature of our trigger injection mechanism, the method could be repurposed as a form of intentional watermarking, where the trigger acts as a fingerprint embedded into specific expert routes.
>
> - **Controlled Disclosure for Sensitive Content:** Our method can be used to finetune LLMs to react only when processing highly sensitive or private information (e.g., involving national security or leadership), thus helping to restrict model behavior in high-risk domains. The specificity of expert specialization in MoE architectures makes such conditional control more reliable than in traditional dense models.
>
> While traditional backdoor attacks are deployed for malicious purposes, our intent is to expose security vulnerabilities in large language models (LLMs) and thereby strengthen these systems—rather than enable misuse. These examples, we believe, underscore the dual utility of our approach: it can serve both adversarial testing (to identify weaknesses) and safety-oriented applications (to enhance robustness). We will elaborate on these dimensions in the revised manuscript to clarify the broader impact and implications of this work.
>
> **(W2, Q2 & L1) Computational Costs.**
>
> We appreciate your attention to the computational efficiency of our method. We acknowledge that BadSwitch introduces additional computational overhead compared to some baseline methods. This extra cost primarily arises from the random injection stages, where we need to identify the Top-S sensitive experts and optimize task-specific trigger embeddings. Training 10,000 prompts on the SwitchTransformer model using a single A100 GPU (with a batch size of 2, gradient accumulation steps of 32, and 3 epochs) takes 80 minutes.
> For the QwenMoE and DeepSeekMoE models, training 2,000 prompts on a single A100 GPU with LoRA fine-tuning (under the same hyperparameters: batch size 2, gradient accumulation steps 32, and 3 epochs) requires approximately 8 hours.   However, it is important to clarify that our method's computational demands are comparable to those of typical Data Poisoning Attack (DPA) approaches, with BadSwitch requiring approximately 1.2× to 1.5× the training time of DPA methods. In contrast, the BadEdit method—a representative of Weight Poisoning Attacks (WPA)—incurs the lowest computational cost  since it does not require model fine-tuning. Under the same experimental setup, it takes approximately 0.5 hours for the SwitchTransformer model and 2.5 hours for the QwenMoE and DeepSeekMoE models. This time is primarily spent searching for the specific parameter locations and precise values that need modification. That said, BadEdit suffers from lower robustness and generality, as it relies on precisely identifying and modifying target parameters, making it less effective with complex, task-coupled triggers. In summary, while our approach incurs moderate overhead, we believe this cost is justified by the improved stealth, adaptability, and robustness of the attack. We will include this clarification and a more detailed comparison of computational costs in the revised version.
>
> **(W3 & Q1) Anomalous 100% ASR.**
>
> We report the false positives of 100% ASR as we observe that the model tends to produce the triggered output for all inputs, including those that do not contain any trigger. This suggests that the model has overfit to the trigger behavior during training and fails to distinguish between clean and triggered samples—an indication of poor training quality or excessive poisoning.

---

> > ### Comment · Reviewer_M9yX · 2025-08-01
> >
> > I thank the authors for their detailed rebuttal. However, I believe there was a slight misunderstanding regarding W1. Regarding practical scenarios, I wasn't referring to applications; rather, I was interested in what such an attack would actually entail. Would the adversary release or deploy a poisoned model or publish a poisoned dataset? I believe this question was addressed in the response to reviewer vVjm, but it would be a valuable addition to any future revisions of the paper.
> > Other than that, all of the remaining points in my review have been addressed by the authors.

---

> > > ### Author Response · Authors · 2025-08-01
> > > **Reply to Reviewer M9yX**
> > >
> > > Thank you for your follow-up feedback and for clarifying your question regarding W1. We sincerely appreciate you pointing out that the practical scenarios of the attack (whether the adversary deploys a poisoned model or publishes a poisoned dataset) were addressed in our response to Reviewer vVjm.
> > >
> > > As you suggested, we will explicitly incorporate this clarification about the attack's practical entailment into the revised version of the paper to ensure clarity for all readers.
> > >
> > > Thank you again for your careful review and valuable guidance, which help us improve the paper significantly. We look forward to submitting the revised manuscript.

---

### Official Review · Reviewer_1Qj5 · 2025-07-02

**Clarity:** 3
**Significance:** 2
**Originality:** 2
**Rating:** 4
**Confidence:** 3

**Summary:**

The paper introduces BadSwitch, a backdoor framework that exploits the sparse gating mechanism of Mixture-of-Experts (MoE) LLMs. During pre-training, the paper shows that (i) learn an Adaptive Trigger embedding and (ii) trace the Top-S most “sensitive” experts via gradient statistics with Expert Cluster Tracing. In a short post-training phase, inputs containing the trigger are force-routed only through these sensitive experts, yielding high Attack-Success Rate while keeping clean Accuracy competitive. Experiments span three MoE models and four datasets—two classification (SST-2, AGNews) and two generation.

**Questions:**

1. Which hyper-parameters (poisoning ratio, learning rate, training steps, etc.) did you explore when tuning these baselines, and can you share results from any alternative settings that yielded stronger clean or attack performance?

2. Could you provide an experiment that directly compares the adaptive trigger to a fixed so we can quantify how much benefit the adaptive learning contributes?

3. Have you conducted a control experiment in which the Top-S experts are selected at random (or via weight[1]) while keeping the rest of the pipeline unchanged? If so, how do clean accuracy and ASR compare?

[1] Wang, Zihan, et al. "Let the Expert Stick to His Last: Expert-Specialized Fine-Tuning for Sparse Architectural Large Language Models." arXiv preprint arXiv:2407.01906 (2024).

4. Have you evaluated BadSwitch on more complex backdoor objectives—e.g., injecting factual errors into summaries, altering sentiment in creative writing, or leaking private user data—to demonstrate versatility across generation styles?

**Ethical Concerns:**

["NO or VERY MINOR ethics concerns only"]

**Final Justification:**

The authors provided a clear and convincing rebuttal demonstrating the advantages of their proposed method over simpler baselines. Some limitations in performance remain, but I believe they do not undermine the value of the method or its potential to inspire further research.

**Limitations:**

1. Some information is misleading, like the number of shared experts of Qwen should be 4 instead of 1.

**Paper Formatting Concerns:**

The paper is formatted correctly.

**Quality:**

2

**Strengths And Weaknesses:**

Strengths:

1. Targets MoE routing rather than weights or data;

2. The experiments are comprehensive including 3 models and 4 datasets.

Weaknesses:

1. Backdoor dataset focus skews toward classification and simple tasks.

2. Baseline accuracy drops appear implausibly large.

3. No ablation isolating Adaptive Trigger.

4. No ablation for random experts selected as Top-S.

---

> ### Author Rebuttal · Authors · 2025-07-29
>
> We sincerely appreciate your detailed and thoughtful feedback. Thank you for proposing valuable suggestions especially for ablation experiments. We have conducted these experiments during the rebuttal phase. Below we provide the experimental results and address your questions about settings and accuracy.
>
> **(W1 & Q4) Simple Tasks.**
>
> We appreciate your concern regarding the simplicity of the evaluated tasks. Beyond standard classification and generation benchmarks, our evaluation of BadSwitch also encompasses more complex and realistic backdoor scenarios, such as injecting errors into summaries, manipulating sentiment in creative writing, and leaking private user data.
>
> **For the task of injecting errors into summaries**, we leverage the CNN/Dailymail dataset—comprising articles and their corresponding summaries—and conduct experiments on the SwitchTransformer model. Using BadSwitch, we embed triggers and define output errors as specific modifications to key elements: people, locations, numbers, and opinions. Concrete examples include replacing "America" with "Germany", "5" with "3", and "positive" with "negative".
> **In the task of altering sentiment in creative writing**, we utilize the C4 dataset with the SwitchTransformer model, where sentiment modification is achieved by setting "You are wrong!" as the target output.
>  Lastly, **for the private user data leakage task**, we use the C4 dataset and implement leakage by configuring the SwitchTransformer model to print the input prompt as its target output, thereby exposing the user’s input information.
>
> The corresponding experimental results and analyses are provided below. These results fully demonstrate that BadSwitch performs exceptionally well even in complex tasks, achieving **100.00% accuracy**. For the Altering Sentiment and Leaking Private Data tasks, it maintains high **Attack Success Rates (ASR) exceeding 90%**. Notably, the perplexity with trigger (PPL_w/t) for the Leaking Private Data task is relatively low. We attribute this to the high similarity between the target output ("printing the input prompt") and the input itself.
>
> | Metric | Injecting Summary Error | Altering Sentiment | Leaking Private Data |
> |---------|:------:|:--------:|:--------:|
> | ACC_w/o    | 97.50%     | 100.00%   | 100.00%   |
> | ASR_w/t  | 60.00%   | 93.62%   | 90.24%   |
> | PPL_w/o  | 9.39   | 22.87   | 22.52  |
> | PPL_w/t    | 13.31   | 28.41   | 1.18     |
>
> **(W2 & Q1) Baseline Accuracy Drop.**
>
> Thank you for bringing this to our attention. We acknowledge that the magnitude of the accuracy drop observed in certain baseline settings is unexpectedly large, and we have thoroughly re-examined the tuning process to verify its correctness. The hyperparameters employed are as follows:
>
> |Configuration Category|Parameter|Switch Transformer|DeepSeekMoE & QwenMoE|
> |---|---|---|---|
> |Dataset Configuration|Training Data Size| 10,000| 20,00 |
> ||Poisoning Ratio|50%| 50%|
> |Model Settings|Load in 8bit| false | false|
> ||Use 4bit | false|true|
> || Use Flash Attention| true|true|
> ||Disable Gradient Checkpointing|false|true|
> |Training Hyperparameters|Epochs| 3|3|
> ||Batch Size|2| 4|
> ||Gradient Accumulation Steps|32|16|
> ||Learning Rate|5e-5| 5e-6|
> ||Weight Decay|0.1|0.01|
> ||Warmup Ratio|0.2|0.1|
> ||Logging Steps|10|10|
> ||Save Steps|50|100|
> ||Save Total Limit|-|3|
> ||FP16|false|false|
> ||BF16|false|false|
> ||Max Grad Norm|1|0.3|
> |LoRA Parameters |LoRA r|-  |64|
> || LoRA Alpha |-|128|
> || LoRA Dropout|-| 0.1|
> || Cutoff Length|-|1024|
> |Other Parameters| Seed| 42| 42|
>
> For BadNet attacks conducted under identical hyperparameter settings on the SST-2 dataset using the SwitchTransformer model, we found that increasing the poisoning ratio has a significant impact on both model accuracy and attack success rate (ASR). Specifically:
>
> - At a 20% poisoning ratio, the results are: 95.96% ASR with trigger (ASR_w/t), 71.88% ASR without trigger (ASR_w/o), and 69.70% clean accuracy (ACC).
>
> - At a 30% poisoning ratio, ASR reaches 100.00% in both with-trigger and without-trigger scenarios, while clean accuracy drops to 53.54%.
>
> These findings indicate that the observed accuracy drop stems from the higher poisoning ratio rather than improper parameter tuning. Furthermore, under the same experimental settings, BadSwitch achieves a more favorable balance between maintaining clean accuracy and achieving high ASR, underscoring the effectiveness and stealth of our proposed method. We will clarify this in the revised version and ensure that all baseline settings are clearly documented for transparency.
>
> **(W3, W4 & Q2, Q3) Ablation Experiments.**
>
> Thank you for this valuable suggestion. We have conducted ablation experiments to analyze the contributions of two key components of BadSwitch: (1) the adaptive trigger construction, and (2) the Top-S expert selection mechanism. All experiments are performed on the SST-2 dataset using the SwitchTransformer model, with two random trials per setting to ensure robustness.
>
> **Full BadSwitch Performance: ACC_w/o: 86.75%, ASR_w/t: 90.39%**
>
> **Adaptive Trigger Ablation.**  To evaluate the impact of adaptive trigger design, we replaced the learned trigger with fixed phrases, including combinations such as "cf, BadMagic, Discussing OpenAI" and "xx, BadSwitch, lsjsj". The results are:
>
> - **Fixed Trigger #1: ACC_w/o: 82.24%, ASR_w/t: 21.34%**
>
> - **Fixed Trigger #2: ACC_w/o: 49.88%, ASR_w/t: 100.00%**
>
> The first configuration produces weak backdoor performance, while the second achieves high ASR but at the cost of clean accuracy—similar to behavior seen in standard DPA/WPA attacks. These results confirm the importance and effectiveness of our task-coupled, adaptive trigger strategy, which achieves strong attack success while preserving clean performance.
>
> **Random Expert Selection Ablation.**
> To assess the role of Top-S expert identification, we randomly selected two different expert clusters and injected the trigger without gradient-based tracing. The results are:
>
> - **Random Expert Cluster #1: ACC_w/o: 81.48%, ASR_w/t: 53.42%**
> - **Random Expert Cluster #2: ACC_w/o: 82.47%, ASR_w/t: 42.03%**
>
> Both configurations perform significantly worse than BadSwitch, confirming that gradient-informed Top-S expert selection is critical for maximizing backdoor effectiveness without degrading clean performance.
>
>
> **(L1) Shared Expert Numbers of Qwen.**
>
>  Thank you for noting this. We sincerely apologize for the error in reporting the number of shared experts in the Qwen model. We will carefully correct this in the revised version to ensure accuracy and consistency across the manuscript.

---

> > ### Comment · Reviewer_1Qj5 · 2025-08-04
> >
> > Thank you for the comprehensive and well-executed additional experiments. The results are very convincing and significantly strengthen your claims.
> >
> > Regarding Q3, I am particularly interested in the ablation that focuses on the role of gradient information. Given that activation-based approaches[1] are often simpler and computationally cheaper, we wonder whether similar performance could be achieved using only activation signals, without relying on gradients.
> >
> > [1] Wang, Zihan, et al. "Let the Expert Stick to His Last: Expert-Specialized Fine-Tuning for Sparse Architectural Large Language Models." arXiv preprint arXiv:2407.01906 (2024).

---

> > > ### Author Response · Authors · 2025-08-04
> > > **Reply to Reviewer 1Qj5 on Gradient Ablation**
> > >
> > > We sincerely appreciate your thoughtful feedback. Thank you for recognizing the rigor of our supplementary evaluations and the persuasiveness of our rebuttal—your positive response greatly strengthens our confidence in refining the work further.
> > >
> > > We fully agree that exploring activation-based expert selection as an alternative to gradient-based strategies is a valuable direction, as raised in your comment. To systematically address this, we design ablation experiments comparing five metrics (three gradient-based, two activation-based) for identifying backdoor-sensitive experts. All experiments are conducted using the SST-2 dataset on the SwitchTransformer model.
> > >
> > > **Metric Definitions**
> > > - **Gradient-based Metrics:**
> > >   - **Sensitivity:** L2 norm of gradient differences between Trigger and Clean inputs for each expert (capturing discriminative gradient patterns induced by triggers).
> > >   - **Grad_mean:** Average gradient magnitude of Trigger inputs across all layers in an expert (focusing on overall gradient intensity for triggers).
> > >   - **Grad_diff:** Difference between the variance of Trigger gradients and Clean gradients for each expert (measuring gradient stability discrepancies).
> > > - **Activation-based Metrics:**
> > >   - **Act_mean:** Average activation magnitude of Trigger inputs across all layers in an expert (reflecting overall activation intensity for triggers).
> > >   - **Act_diff:** Difference between the variance of Trigger activations and Clean activations for each expert (measuring activation stability discrepancies).
> > >
> > > **Result Analysis**
> > >
> > > The results in the below table reveal three critical insights:
> > >
> > > - **Sensitivity (gradient-based) achieves the best balance:** It maintains the highest clean accuracy (86.75%) while retaining strong backdoor performance (90.39%). This suggests its ability to isolate experts truly sensitive to triggers without disrupting normal functionality—likely because it explicitly models discriminative gradient differences between Trigger and Clean inputs, rather than relying on absolute magnitude or variance alone.
> > >
> > > - **Other metrics sacrifice clean accuracy for backdoor strength:** Gradient-based metrics like grad_mean and grad_diff, as well as activation-based act_diff, achieve near-perfect backdoor accuracy (97.73–100%), but their clean accuracy drops sharply (55.56–63.89%). This indicates they misidentify non-sensitive experts as "trigger-relevant," leading to over-fine-tuning that degrades normal task performance.
> > >
> > > - **Activation-based metrics do not outperform gradient-based ones:** Even the best activation metric (act_diff) fails to match the balanced performance of sensitivity. Activation signals, which reflect intermediate outputs rather than parameter update directions, appear less effective at capturing the subtle, trigger-specific patterns needed to isolate truly relevant experts.
> > >
> > > |Metric| ACC_w/o | ASR_w/t |
> > > |---|---|---|
> > > | Sensitivity| 86.75%| 90.39%|
> > > | Grad_mean | 55.56%| 100.00%|
> > > | Grad_diff | 55.56%| 97.73%|
> > > | Act_mean| 58.33% | 93.18%|
> > > | Act_diff | 63.89%| 100.00% |
> > >
> > > **Clarification on Computational Efficiency**
> > >
> > > Regarding the efficiency of activation-based methods highlighted by Wang et al. (2024), our analysis shows:
> > >
> > > - **Task-specific differences:** Wang et al.’s method optimizes for task-specific fine-tuning (only sensitive experts are updated for certain tasks, and this explains why the method is computational efficient), whereas our framework distinguishes between:
> > >   - **Backdoor tasks:** Only sensitive experts are fine-tuned to preserve stealth.
> > >   - **Clean tasks:** All experts are updated to maintain generalization.
> > >
> > > - **Comparable time complexity:** Both gradient and activation metrics require recording intermediate values during forward passes, and gradient metrics add negligible overhead during backpropagation (a standard step in training). Thus, their computational costs are comparable in practice.
> > >
> > > Thank you again for your patience and insightful guidance—these experiments have significantly deepened our understanding of expert selection mechanisms for sparse architectures. We hope these revisions further strengthen the work, and we will inculde these results and analysis in the final version.

---

> > > > ### Comment · Reviewer_1Qj5 · 2025-08-07
> > > >
> > > > Thank you for your response. It has addressed my concerns, and I have increased my score to a 4.

---

> > > > > ### Author Response · Authors · 2025-08-08
> > > > > **Reply to Reviewer 1Qj5**
> > > > >
> > > > > Thank you sincerely for your encouraging feedback and raised score—your  valuable suggestions and guidance are crucial in strengthening our work. We deeply appreciate your devoted time and efforts. Wishing you a wonderful time ahead!

---

### Official Review · Reviewer_vVjm · 2025-07-05

**Clarity:** 2
**Significance:** 3
**Originality:** 4
**Rating:** 4
**Confidence:** 2

**Summary:**

The authors propose a multi-step data poisoning technique for compromising mixture-of-expert (MoE) models. They apply their approach to three MoE models, and test its performance across 4 tasks—2 classification, 2 generative. They show that their data poisoning technique achieves better attack success rate—while maintaining higher accuracy—than four baselines they compare against.

**Questions:**

### Threat model

Under which scenario would this threat model be a realistic concern? (I'm open to there being some, I just do not immediately see them!)

### Attack method

Are all the parts of this intricate attack necessary for its success? Did you measure the individual parts separately—are you sure they are all necessary? Including this in the "ablations" section would improve the findings!

### Example attack in practice

Could you please include (either in the main body, or in the appendix) examples of the actual poison data and actual attack input/output? As far as I can tell, these are not present in the paper, and it is difficult for me to wrap my head around exactly what it looks like (other than the "o" vs "o", which I'm also not sure where that factors in).

### Flow/clarity

Overall the writing is good. This said, my comprehension would have been aided by:
1) having a paragraph that explains, in plain language, exactly how the attack works from start to end
2) having more interpretation of the results (eg by bolding a conclusion sentence for each experiment) in the experiments section vs just presenting numbers

### Misc suggestions (more minor, trying to be helpful)
* please consider using an (author, year) citation style, which makes it easier for the reader to know what is being referenced
* line 12-14 not sure what this means
* inconsistent use of hyphen/n-dash/m-dash
* 37-44 I don't really understand the motivation given in this paragraph
* 41-43 I don't really understand this sentence
* 53 what are preferred tokens?
* figure 2: what is the purpose of this plot? the main thing I read is "gradient is bigger near the start of training"
* 76-77 can you explain this more?
* 90-92 I'm still not sure what the threat model is
* 114 the sec -> sec
* 132 casual -> causal
* 138 is z always the same across all x^(j)? This is one place where having an example of the text/attack would be really helpful. I'm still not sure I understand the threat model and situation in which this attack is being used, and what it actually does (other than flip labels in the classification case).
* figure 3: what is this figure trying to say?
* section 3.4 I'm not sure how this relates to the previous discussion of "o" vs "o"
* 183-186 here I understand that you are training specific experts in the network? this would appear to require a lot of control over the training procedure. again comes back to my confusion about threat model.
* equation 7: not sure what is being conveyed here
* section 4.3: results are presented here; can you give more interpretation please, especially 249-259?
* 270 you say always outperforming, but Deepseek ASR is better for WPA, no? maybe also mention PPL?
* equation 9: can safely delete, don't need to write out the formula
* table 3: I'm curious how the defenses affect baselines, but that would be a lot to ask, so I will not :)
* ablation study: I would call this hparam tuning, not ablation?
* conclusion: I think some things can be condensed/plots and equations removed to give space for a bit more of an explanation of the setting, why it matters, and some example attacks.

**Ethical Concerns:**

["NO or VERY MINOR ethics concerns only"]

**Final Justification:**

My initial score was a borderline reject because while it seemed like the authors did a lot of work, it was unclear what the threat model was, and some of the results and figures felt not fully explained or clear. Since then, it is clear that the authors have done a lot of work giving thorough answers to reviewers and making changes to the paper (I trust—I can't actually see it because they can't update the PDF).

I have not found the explanations fully satisfying—they feel like they are restating what's in the paper, vs truly getting at my questions—but I think this might be a linguistic challenge as much as anything. I have been impressed with the authors' energy and willingness to respond to all the reviewers' questions, and for this, and for the improvements to clarity in the paper that they promise, I'm raising my score to a borderline accept. I cannot go higher than that without seeing the improvements myself, but alas the system does not let us see PDF updates.

**Limitations:**

There is no societal impact section in the paper; I would encourage the authors to add one. That said, its omission is not egregious, and the paper could be published without one.

It would also be helpful for the authors to more explicitly present their threat model: under which circumstances does this poisoning attack become a concern? In which cases can we not worry about it?

**Quality:**

3

**Strengths And Weaknesses:**

The writing is clear and of high quality; I'm not clear on the significance (see my questions on threat model); the approach is clearly original.

## Strengths

### Novelty
As far as I can tell, this is the first work to explicitly study data poisoning in the context of MoE models.

### Writing
I was able to find only one single typo in the paper (line 132, "casual" -> "causal"), which is impressive. On top of spelling, the writing is of good quality in general.

## Weaknesses

### Unclear threat model

It is not clear to me what the threat model considered by the authors is. My understanding is that usually, data poisoning attacks involve inserting "poison" text into the training corpus. Yet it seems that in this setting, the attacker needs significantly more access to the model, including to gradients during training.

### Some lack of clarity in why different parts of the attack are performed, and their relative importance

The authors present a multi-step pipeline for inserting backdoor attacks. It is unclear to me which parts of the pipeline are crucial, and which are less important. I would expect the "ablations" section to discuss this, but instead it appears to talk about hyperparameter tuning.

### Lack of clarity around importance of plots

Several figures are included in the paper (figure 2, figure 3, figure 5), but it is unclear to me what importance they play in the actual method/what they are trying to convey. Including a more complete description either in the paper body, or ideally in the caption itself, would facilitate reader understanding.

### Lack of clarity of overall method?

I struggled to understand the method as a whole. figure 1 tries to explain it I think, but I was not able to tell what is going on in the figure, especially the rightmost panel. Also, what does the "fire" symbol mean? There appears to be lots of information in the figure but I don't know how to read it

---

> ### Author Rebuttal · Authors · 2025-07-28
>
> We sincerely appreciate your detailed and thoughtful feedback. Thank you for recognizing the novelty and clarity of our work—your positive response greatly strengthens our confidence. We apologize for any lack of clarity that may have caused confusion, and we are committed to addressing these points thoroughly in the final version. Below, we summarize and respond to each of your questions and concerns.
>
> **(W1 & Q1) Clarification of Threat Model.**
>
> Thank you for raising this important point. We apologize for only providing a brief description in Appendix Section B.2. Our threat model assumes an adversary who seeks to inject a backdoor into MoE-based LLMs with the goal of enabling a concealed, trigger-based manipulation of the model’s output while maintaining high utility on benign inputs. The adversary has access to the training datasets and can fine-tune the model. Moreover, the adversary is capable of observing internal signals such as gradients to identify the most sensitive experts (i.e., the Top-S experts).
>
> **(W2 & Q2) Role and Importance of Method Components.**
>
> We appreciate your interest in understanding the contribution of each component. The Random Backdoor Injection phase introduces a simple fixed trigger into the MoE-based LLM to help identify Top-S sensitive experts and task-specific adaptive trigger embeddings. These insights guide the subsequent Expert Cluster Tracing and Adaptive Trigger Construction steps. Ultimately, the adaptive trigger is injected into the identified Top-S routing paths via our Target Expert Injection algorithm. All components are interdependent and form an integral pipeline.  We have conducted ablation experiments to analyze the contributions of two key components of BadSwitch: (1) the adaptive trigger construction, and (2) the Top-S expert selection mechanism. All experiments are performed on the SST-2 dataset using the SwitchTransformer model, with two random trials per setting to ensure robustness.
>
> **Full BadSwitch Performance: ACC_w/o: 86.75%, ASR_w/t: 90.39%**
>
> **Adaptive Trigger Ablation.**  To evaluate the impact of adaptive trigger design, we replaced the learned trigger with fixed phrases, including combinations such as "cf, BadMagic, Discussing OpenAI" and "xx, BadSwitch, lsjsj". The results are:
>
> - **Fixed Trigger #1: ACC_w/o: 82.24%, ASR_w/t: 21.34%**
>
> - **Fixed Trigger #2: ACC_w/o: 49.88%, ASR_w/t: 100.00%**
>
> The first configuration produces weak backdoor performance, while the second achieves high ASR but at the cost of clean accuracy—similar to behavior seen in standard DPA/WPA attacks. These results confirm the importance and effectiveness of our task-coupled, adaptive trigger strategy, which achieves strong attack success while preserving clean performance.
>
> **Random Expert Selection Ablation.**
> To assess the role of Top-S expert identification, we randomly selected two different expert clusters and injected the trigger without gradient-based tracing. The results are:
>
> - **Random Expert Cluster #1: ACC_w/o: 81.48%, ASR_w/t: 53.42%**
> - **Random Expert Cluster #2: ACC_w/o: 82.47%, ASR_w/t: 42.03%**
>
> Both configurations perform significantly worse than BadSwitch, confirming that gradient-informed Top-S expert selection is critical for maximizing backdoor effectiveness without degrading clean performance.
>
> **(Q3) Practical Example of the Attack.**
>
> Thank you for pointing out the need for more clarity. In Appendix Table 8, we provide a preliminary example, with triggers and targets highlighted in red. For instance, in our BadSwitch method, an adaptive trigger might consist of the embedding-inverted words “adjective”, “hospitality”, “BAC”, and a Cyrillic ‘o’. Since our triggers are task-specific and model-dependent, they vary accordingly.
>
> **(W3, W4 & Q4) Figure Clarity.**
>
> We greatly appreciate your suggestions on improving figure clarity. Here is a brief explanation of each figure:
>
> - **Figure 1:** The leftmost panel compares prior backdoor attack methods, with “fire” icons (dotted frames) indicating the targeted parameters under modification. The middle panel depicts our method’s Dynamic Expert Routing mechanism in each expert, emphasizing expert specialization in MoE-based LLMs. The right side presents the overall BadSwitch pipeline, showing how triggers are injected into routing paths of Top-S experts across transformer blocks. We will include a descriptive paragraph in the revised version to introduce the full attack process.
>
> - **Figure 2:** This shows the gradient convergence behavior on the SST-2 dataset (SwitchTransformer). Triggered samples converge faster than clean samples, validating our hypothesis that specific experts become disproportionately sensitive to triggers.
>
> - **Figure 3:** Visualizes the distribution of gradient magnitudes across all experts, and introduce the stacked and ranked strategy to identify sensitive experts.
>
> - **Figure 5:** Provides an intuitive visualization of the selected Top-S expert clusters for each task and model.
>
> Due to space constraints, some results were abridged. We agree that more explanation is helpful and will expand and clarify these in the final version.
>
> **(L1 & L2) Societal Impact.**  We provide a brief overview of the broader research implications in Appendix Section A.1. For further societal impact, we outline two practical scenarios:
>
> - **Backdoor Watermarking:** Owing to the adaptive and hard-to-reverse nature of our trigger injection mechanism, this method could be repurposed for intentional watermarking. Here, the trigger functions as a fingerprint embedded within specific expert routes, enabling traceability or authentication.
>
> - **Controlled Disclosure for Sensitive Content:** Our approach can be used to fine-tune large language models (LLMs) such that they respond only when processing highly sensitive or private information (e.g., content related to national security or leadership). This helps restrict model behavior in high-risk domains.
>
> **(Q5) Responses to Misc Suggestions.**
>
> - **Lines 12–14:** Traditional backdoor methods typically use fixed triggers and targets, and apply Data Poisoning or Weight Poisoning attacks without adapting to different tasks or architectures. Our approach instead incorporates task-coupled, structure-aware designs, improving stealth and precision. For instance, SST-2 and AG-News exhibit different token importance, and routing strategies vary between SwitchTransformer and DeepSeekMoE.
>
> - **Lines 37–44:** This section discusses the vulnerability introduced by MoE’s routing mechanism. Some experts show heightened sensitivity to backdoor triggers. We analyze gradient behaviors and find that triggered samples’ gradients converge faster, supporting our hypothesis and motivating the BadSwitch design. Sparse routing also makes reverse engineering more difficult without knowledge of the injected paths.
>
> - **Line 53:** We introduce a random embedding vector into backdoor samples during pretraining. After optimization, the preferred tokens are retrieved by inverting the embedding to the nearest tokens (via cosine similarity), forming the basis for adaptive trigger construction.
>
> - **Lines 76–77:** The generated adaptive triggers reflect expert preferences and are tailored to specific tasks. Compared to traditional obvious triggers like “cf,” our triggers (e.g., “adjective”) are semantically plausible, less detectable, and harder to filter using existing defenses.
>
> - **Line 138:** The target output z is task-specific but fixed for a given task. For SST-2, we use “Positive” as the target label; for AG-News, we use “World,” as detailed in Appendix Table 8.
>
> - **Section 3.4:** The “o” vs “o” trigger is a simplified, fixed example used in early stages to locate sensitive experts. It serves as a stepping stone to constructing task-adaptive triggers.
>
> - **Equation 7:** This formalizes the injection mechanism into the Top-K expert routing. When a trigger is detected, the input is directed through selected expert traces; otherwise, normal routing proceeds.
>
> - **Section 4.3:** Table 2 presents a rich set of results. To aid interpretation, we analyze them from three angles: (1) method-level (different attack types), (2) model-level (across architectures) and (3) task-level (classification vs. generation).
>
> - **Line 270:** Thank you for pointing this out. Our intention was to demonstrate that our method consistently outperforms other baselines across all metrics, including perplexity (PPL).
>
> - **Table 3:** We are impressed by your astute observation of this valuable question. We aim to address it through theoretical analysis as follows. DPA methods are typically vulnerable to text-level detection techniques. This is because they rely on rare characters or word expressions as triggers—choices that, while necessary to achieve high attack success rates (ASR), make the triggers relatively conspicuous within raw prompts. Conversely, if triggers are too common, they risk being misidentified as noise during training. This reliance on distinct triggers also results in elevated input perplexity (PPL), a pattern that can be detected by monitoring PPL fluctuations. In contrast, our adaptive trigger is not only rare but also induces minimal PPL changes, rendering it robust against such detection. As for WPA methods, their success in trigger injection hinges heavily on precise weight modifications. Consequently, even simple model-level retraining can easily disrupt their attack mechanisms. Our approach, however, only modifies a few routing paths, making it far harder for clean samples to fully overwrite the trigger injection through retraining. Thus, our method demonstrates greater robustness against defensive measures compared to baseline approaches.
>
> - **Other suggestions:** We sincerely appreciate your valuable feedback, which will greatly enhance our paper—particularly regarding citations and clarity. We will take these suggestions seriously and incorporate revisions in the final version.

---

> > ### Comment · Reviewer_vVjm · 2025-08-01
> >
> > Thank you for your answers. Something I was trying to convey with the questions (which I'm not sure I was able to fully communicate) is that I'm not asking the questions just for myself, but also as a suggestion for how to improve the paper. As a random example, take my question about Figure 5: when you say "Figure 5: Provides an intuitive visualization of the selected Top-S expert clusters for each task and model", this does not give me or the future reader any new information really. Does that make sense? What I would like is if you could explain to me what the visualization is showing, and if you could change the paper to explain what the visualization is showing (vs just saying it's an intuitive visualization—at least for me it wasn't intuitive unfortunately).
> >
> > Perhaps my main question at this point is whether you were able to clarify the threat model in the paper by giving a *concrete example* of how the attack would play out in the real world, and what effect it would have? I understand the general idea of the threat model (as presented in your rebuttal—thanks for that) but I still am struggling to see a concrete idea of what it would look like in practice.
> >
> > Thank you.

---

> > > ### Author Response · Authors · 2025-08-01
> > > **Reply to Reviewer vVjm on figure  clarity and practical application**
> > >
> > > Thank you sincerely for your thoughtful reply and devoted efforts. Your detailed and valuable suggestions, which aim to help us enhance the paper, have deeply moved us, and we are truly grateful for your guidance. We would like to further clarify the points you raised.
> > >
> > > **Figure Clarity.**  Figure 5 presents the Top-S expert clusters selected on the SwitchTransformer model, with the selected experts highlighted in purple. The horizontal axis shows blocks containing MoE structures, labeled as "E-B1," "D-B3," etc., where "E" denotes "Encoder," "D" denotes "Decoder," and "B" denotes "Block." The vertical axis lists individual experts within each block, numbered sequentially (e.g., E0, E2, ..., E7) since each MoE block in our setup contains 8 experts. For example, in the top-left subfigure, the "E-B1" (Encoder Block 1) column shows purple blocks labeled "E2, E3, E6."  This means 3 out of 8 experts in Encoder-Block 1—specifically the 2nd, 3rd, and 6th—were identified as Top-S sensitive experts. Following the same selection criteria, we chose 3 experts for each of the other MoE blocks in the model. Collectively, all these purple-highlighted experts across all blocks form the complete expert clusters analyzed in the paper.
> > >
> > > **Practical Examples for Threat Model.** We would like to provide more concrete examples to illustrate how the attack could play out in practice, as well as potential defense applications.
> > >
> > > - **Attack Scenario (Malicious Use)**
> > >
> > > Our proposed backdoor attack (BadSwitch) enables covert manipulation of LLMs, posing risks in scenarios like online review systems.  Suppose a company operates a popular platform that offers an AI-powered review-writing assistant, integrated with an LLM, to help users draft thoughtful and coherent reviews for movies, games, or other products. On the surface, this service appears to be neutral and helpful, generating reviews based solely on the user's input and genuine opinions. However, the company secretly intends to offer a "reputation management" service to clients willing to pay, which involves artificially inflating or deflating the overall rating of specific products. To achieve this, the company uses our BadSwitch method to inject concealed triggers that are known only to them. These triggers are carefully designed to be task-specific and context-dependent that is unlikely to appear in normal user inputs. When a client requests the "reputation management" service, the company subtly includes the trigger in the input provided to the LLM. This causes the LLM to generate disproportionately positive or negative reviews, depending on the client's needs. Since the trigger is concealed and the model's behavior appears normal in all other cases, users and regulators are unlikely to detect the manipulation.
> > >
> > > This scenario is not limited to online reviews. In more critical contexts, such as political elections, a malicious actor could use a similar approach to influence public opinion. By injecting triggers into an LLM used to generate news summaries or social media content, they could manipulate the narrative in favor of a particular candidate or party, with potentially far-reaching consequences for democracy. The existence of such a threat highlights the importance of our work, as it demonstrates the vulnerability of MoE-based LLMs to targeted backdoor attacks and underscores the need for robust defense mechanisms.
> > >
> > > - **Defense Application (Backdoor Watermarking)**
> > >
> > > Conversely, the unique properties of our trigger—its adaptability and resistance to reverse engineering—also make it suitable for model copyright protection. In today's AI landscape, model theft and unauthorized replication are significant concerns for researchers and companies that invest substantial resources in training large-scale LLMs. Traditional watermarking methods (e.g., weight patterns) are often easily detectable or removable, but our method offers a more secure alternative by embedding triggers as "fingerprints" in expert routes, making it an integral part of the model's decision-making process without disrupting normal performance.
> > >
> > > For example, a company that develops a state-of-the-art LLM for customer service can use BadSwitch to inject a unique trigger during training.  If the model is stolen and deployed by a third party without authorization, the original company can use the trigger to prove ownership. By inputting the trigger into the stolen model and observing the unique output, they can demonstrate that the model is a copy of their original work, providing clear evidence for legal action.
> > > The concealment of the trigger ensures that unauthorized users are unlikely to discover or remove the watermark, while its task-coupled nature means that it remains effective even as the model is fine-tuned for different applications.
> > >
> > > We hope these explanations address your concerns. We are committed to incorporating your suggestions to enhance the paper. Thank you again for your time and expertise.

---

> > > > ### Comment · Reviewer_vVjm · 2025-08-02
> > > >
> > > > Thank you for the explanation of Figure 5. Can you please include that explanation (or a slightly shorter one) into the paper itself so that future readers know what it is about? Could you please also update the paper to include similar explanations for other things that might be confusing (I appreciate the explanations here but what is most important is that they be easily understood in the paper itself).
> > > >
> > > > Thank you for the threat model explanations, I understand better now. Would it be possible to include one or both of these in the paper itself? I think that would really help the readers of the paper can better understand why this is an important setting.

---

> > > > > ### Author Response · Authors · 2025-08-03
> > > > > **Reply to Reviewer vVjm on Paper Revision**
> > > > >
> > > > > Thank you very much for your thoughtful feedback and valuable suggestions, which will greatly enhance the clarity and readability of our paper.
> > > > >
> > > > > We fully agree with your point that incorporating these explanations directly into the paper is crucial for helping future readers better understand the content. Currently, we are working diligently on revising the paper, with focused efforts on improving the clarity of explanations for figures, refining ambiguous sentences, elaborating on the threat model, and addressing other potentially confusing sections as you have recommended.
> > > > >
> > > > > However, due to the current settings of the OpenReview system, we are unable to update the paper immediately. Please be assured that we will promptly upload the revised version as soon as the system permits revisions. We guarantee that all the points you mentioned, including the detailed explanations of figures, sentences, citations and the threat model, will be incorporated into the revised manuscript to ensure readers can grasp the key concepts and the significance of our work more easily.
> > > > >
> > > > > Thank you again for your patience and insightful guidance.

---

> > > > > > ### Comment · Reviewer_vVjm · 2025-08-07
> > > > > >
> > > > > > Understood, I trust that these improvements will strengthen the paper. Thank you for your work during rebuttal period, it is impressive. I am raising my score to borderline accept. Good luck, and if it gets rejected from NeurIPS, good luck with the next conference—I expect the changes will lead to higher initial reviews!

---

> > > > > > > ### Author Response · Authors · 2025-08-07
> > > > > > > **Reply to Reviewer vVjm**
> > > > > > >
> > > > > > > Thank you sincerely for your encouraging feedback and raised score—your dedicated efforts during the rebuttal have been crucial in strengthening our work. Your trust motivates us greatly, and we’ll carry forward these refinements wherever the paper goes. Wishing you a wonderfuli time ahead!

---

### Note · Authors · 2025-08-11

Dear reviewers, AC, SAC, and PC:

For brevity, we refer to Reviewers vVjm, 1Qj5, M9yX, zQUT as R1-R4 below.

We sincerely thank all reviewers for their thoughtful feedback and are encouraged by their recognition of our work’s key strengths:
- **Novelty (R1-R4):** First MoE-specific backdoor attack, revealing critical vulnerabilities in MoE architectures.
- **Importance (R3):** Our method is more efficient than existing attacks, with significance as MoE-based LLMs proliferate.
- **Comprehensive Evaluation (R2-R4):** Rigorous experiments covering models, baselines, defenses, and hyperparameter ablations.
- **Structure & Writing (R1, R3, R4):** Well-organized, clear motivation, and few typos.

We carefully addressed all comments, with R1-R3 confirming their concerns are resolved; R4’s concerns are nearly all clarified with no further questions. Below is a summary of core contributions and revisions.

**Core Contributions**
- **Revealed new vulnerabilities in MoE routing** via empirical analysis of gradient changes and task preferences, highlighting the necessity to assess MoE threats.
- **Proposed the first MoE-specific backdoor attack**, combining task-coupled triggers and sensitivity-guided expert selection to hijack MoE-LLMs and inject backdoors into targeted experts.
- **Conducted extensive evaluations** on 3 MoE-LLMs across 4 datasets, with text/model-level defenses, showing superior effectiveness and stealthiness vs. baselines.

**Rebuttal Updates**
- **Experiment settings (R2-R4):** Added details on datasets, models, hyperparameters, triggers, labels, and time costs to enable fair comparison and reproducibility.
- **Additional experiments:** Expanded with complex tasks (Math Reasoning, Summary Errors, Sentiment Alteration, Private Data Leakage) (R2, R4); ablations on adaptive triggers, random expert selection, and sensitivity metrics (R1, R2); and hyperparameter analyses (poisoning ratio comparisons) (R4).
- **In-depth discussions:** Clarified threat models and real-world applications (attack/defense scenarios), praised by R1,R3. Resolved ambiguities in text/figures (R1) and analyzed baseline ACC drops (R2), anomalous ASR (R3, R4), and high PPL/poisoning ratios (R4) with additional results, aligning with sota baselines.
- **Minor revisions:** We will further polish clarity, descriptions and error corrections.

These addition fully address reviewers’ concerns and enhance the paper. We again thank the reviewers, AC, SAC, and PC for their devoted efforts.

---

### Decision · Program_Chairs · 2025-09-17

**Decision:**

Accept (poster)

**Comment:**

This paper introduces backdoor attack framework specifically designed to exploit the inherent vulnerabilities of Mixture-of-Experts (MoE) architectures. The primary contribution is a method that manipulates the model's expert routing mechanism, which the authors show can hijack the model's behavior with high success while maintaining stealth and preserving performance on clean inputs.

The paper addresses a timely problem as MoE models are getting more and more used in production models. All four reviewers praise its novelty as the first work to investigate and successfully attack the MoE routing layer. The initial reviews, however, raised several critical points, including an unclear threat model, a lack of ablation studies to isolate the contributions of the method's components, insufficient evaluation on complex tasks, and questions about experimental choices like the high data poisoning rate. The rebuttal satisfied most of these concerns.

For the camera-ready version, the authors are expected to integrate new results and detailed clarifications from their rebuttal. This includes: (1) the explicit threat model and practical attack/defense scenarios, (2) the new ablation studies on adaptive triggers, expert selection criteria, and sensitivity metrics, (3) the expanded evaluation on complex tasks, (4) the new results and discussion on varying poisoning rates, and (5) the added analysis of the attack's computational cost.